# *Jouvence* a small nucleolar RNA required in the gut extends lifespan in *Drosophila*

Stéphanie Soulé [1,4], Lucille Mellottée [1,4], Abdelkrim Arab[1], Chongjian Chen[2,3] & Jean-René Martin [1✉]

Longevity is influenced by genetic and environmental factors, but the underlying mechanisms remain elusive. Here, we functionally characterise a *Drosophila* small nucleolar RNA (snoRNA), named *jouvence* whose loss of function reduces lifespan. The genomic region of *jouvence* rescues the longevity in mutant, while its overexpression in wild-type increases lifespan. *Jouvence* is required in enterocytes. In mutant, the epithelium of the gut presents more hyperplasia, while the overexpression of *jouvence* prevents it. Molecularly, the mutant lack pseudouridylation on 18S and 28S-rRNA, a function rescued by targeted expression of *jouvence* in the gut. A transcriptomic analysis performed from the gut reveals that several genes are either up- or down-regulated, while restoring the mRNA level of two genes (*ninaD* or CG6296) rescue the longevity. Since snoRNAs are structurally and functionally well conserved throughout evolution, we identified putative *jouvence* orthologue in mammals including humans, suggesting that its function in longevity could be conserved.

[1] Institut des Neurosciences Paris-Saclay (Neuro-PSI), UMR-9197, Equipe: Imagerie Cérébrale Fonctionnelle et Comportements (ICFC), CNRS/Université Paris Sud, 1 Avenue de la Terrasse (Bat. 32/33), 91198 Gif-sur-Yvette, France. [2] Genetics and Developmental Biology Department, Mammalian Developmental Epigenetics Group (E. Heard Lab.), Institut Curie, CNRS UMR-3215, U934 Inserm, 26 rue d'Ulm, Paris 75005, France. [3] Present address: Corporate HQ: Building B2, Yard 88, kechuang 6 Rd, Beijing Economic-Technological Development Area, 100176 Beijing, China. [4] These authors contributed equally: Stéphanie Soulé, Lucille Mellottée. ✉email: jean-rene.martin@inaf.cnrs-gif.fr

The aging process is the result of complex biological mechanisms leading to accumulation of different types of damage at molecular, cellular, tissue and organ levels. This leads to the decrease or loss of certain physiological functions, and therefore, to an increase of vulnerability to diseases and death[1,2]. Several genes and metabolic factors have been reported to modulate longevity[3–16]. Indeed, genes that increase longevity have been identified in different model organisms, such as yeast[3,4], *C. elegans*[17], *Drosophila*[5–16] and mouse[18]. More particularly, in *Drosophila*, these include *mathuselah*[5], insulin-signalling pathway genes[6,7,15], Cu/Zn superdismutase[9], sirtuin[3,4,10], and genes affecting either the regulation of mitochondria as PGC-1[11] or the mitochondrial respiratory chain[12]. It is also well accepted that lifespan is environmentally modulated. In addition, longevity positively correlates with the ability to resist to stress. In such way, environmental factors like diet[1], and stress[5,14] also modulate an organism's lifespan. More recently, some microRNAs (miRNAs) were identified as key regulators of genes involved among other processes as neurodegeneration, in aging[19–21].

The snoRNAs are members of a group of non-coding RNAs. SnoRNAs are found in species from archeobacteria (in which they are named sRNAs) to mammals[22]. They are known to be present in the nucleolus, in which they are associated with a set of proteins to form small nucleolar ribonucleoprotein (snoRNPs)[22]. They are generally known to be processed from introns of pre-mRNAs. They are believed to have several functions, including the 2'-O-methylation and pseudouridylation of different classes of RNA, the nucleolytic process of the rRNA, telomeric DNA synthesis[22,23], as well as they can target the spliceosomal snRNAs (whose are their main targets after rRNAs). Based on conserved secondary structure and functional RNA motifs, two major classes have been distinguished, the box C/D and box H/ACA. The box C/D, comprising notably the multiple "U" snoRNAs (although not all "U" are necessarily snoRNAs), generally performs the 2'-O-methylation of ribosomal RNA, while the box H/ACA directs the conversion of uridine to pseudouridine, and so particularly of rRNA[22,23]. However, recent studies have demonstrated that the H/ACA snoRNA might also pseudouridinylate other RNA substrates, such as mRNA and long-non-coding-RNAs (lncRNAs)[24] or have a role in chromatin remodelling, indicating that they have several other functions[25]. Then, although few H/ACA box snoRNA have been involved in human diseases[26,27], their role at the organismal level and more particularly in aging has not yet been documented.

Here, we characterize a small nucleolar RNA (snoRNA:Ψ28S-1153) that we name *jouvence* (*jou*) and show that its deletion reduces lifespan. A transgene containing the genomic region of *jou* is able to rescue the lifespan, while its overexpression increases it. In situ hybridization (ISH) reveals that *jou* is expressed in the enterocytes, the main cell-types of the epithelium of the gut. At the cellular level, in *jou*-deleted flies, the number of enterocytes as well as the proliferative cells increase leading to an intestinal hyperplasia. Genetic targeted expression of *jou* in the enterocytes prevents the hyperplasia, which is sufficient to rescue and significantly increase longevity, and so, even when it is expressed only in adulthood (through Gene-Switch conditional expression). Finally, as snoRNAs are generally well conserved throughout evolution, both structurally and functionally, we identify putative *jou* mammalian orthologues, both in mouse and human, suggesting that it may have an implication in mammalian aging.

## Results

**Genetic and molecular characterization of the snoRNA jouvence**. Aging involves a progressive decline and alteration in tissues and sensory-motor functions[28–30]. In a study to characterize

putative genes involved in locomotor activity and more particularly genes expressed in the central complex (CC), a premotor centre localised in the middle of the brain[31], a P-element insertional mutagenesis was performed. We identified the enhancer-trap line P[GAL4]4C, and show that the blockage of the targeted ring neurons disrupts the power law distribution of walking time intervals[32]. Here, in a step further, we molecularly characterize the P[Gal4]4C line, as well as we study it in the context of aging and particularly in aging effects on sensory-motor performance. P[GAL4]4C is inserted on the second chromosome, at position 50B1 (Fig. 1a), between two putative genes: CG13333 and CG13334. CG13333 is embryonically expressed[33], while CG13334 shows a partial and weak homology to the lactate dehydrogenase gene. Furthermore, a bioinformatic study[34] has annotated a snoRNA (snoRNA:Ψ28S-1153) between CG13333 and CG13334 near the P[GAL4]4C insertion (Fig. 1a), while a recent developmental transcriptomic analysis has annotated two additional putative snoRNAs (snoR-NA:2R:9445205 and snoRNA:2R:9445410) (http://flybase.org/)[35] localized just upstream to the first snoRNA:Ψ28S-1153. However, according to the canonical definition of snoRNA in which the H/ACA box is predicted to form a hairpin-hinge-hairpin-tail structure (Fig. 1d), while the C/D box is predicted to form a single hairpin[22,23], these two last snoRNAs do not fulfil these two criteria, suggesting that they are likely not true or, at least canonical snoRNAs. Whereas the snoRNA:Ψ28S-1153 forms a typical H/ACA double hairpin (Fig. 1e). To generate mutations in this complex locus, the P[GAL4]4C was excised, using the standard genetic method of excision[36] resulting in a small 632 bp deletion (F4) (shortly named: Del) (Fig. 1a and Supplementary Fig. 1a), which completely removed the snoRNA:Ψ28S-1153 as well as the other two RNAs. Next, we performed RT-PCR to test if this deletion had an effect on the expression of the two encoding neighbouring genes. Both genes were normally expressed in the deletion line compared to control Wild-Type (WT) Canton-S (CS) flies (Supplementary Fig. 1b).

**Genomic deletion encompassing the snoRNA reduces lifespan.** Interestingly, we observed by daily husbandry, that the deletion flies had a shorter lifespan. Hence we compared the longevity of these flies to that of Control CS. Since small variations in the genomic background can affect longevity, we outcrossed the deletion line a minimum of 6 times to CS (Cantonisation). Longevity tests (Fig. 1b) performed on females revealed that deletion flies lived shorter compared to control. To test the snoRNA's role in the longevity effect, three transgenic lines containing a DNA-fragment of 1723 bp, comprising the putative genomic and regulatory region of the snoRNAs (genomic-rescued named: rescue-1, 2 and 3) (Fig. 1a: red bar) were generated, and tested for longevity against the deletion background. Again here, these transgenic lines were outcrossed 6 times to CS and then crossed to the deletion to generate the Del+rescue-1, -2, -3 lines respectively. Longevity in Del+rescue-1 females was rescued, while lifespan in the rescue-1 line itself increased (Fig. 1b) (see Supplementary Fig. 1d for the replicate of this longevity experiment). Similar results were also observed with the two other independent insertions: rescue-2 and rescue-3 (Fig. 1c). In addition, as aforementioned, since longevity is very sensitive to the genetic background, we also tested the phenotype of the deletion in an independent genetic background (Berlin)[37]. The deletion flies in a Berlin genetic background still live shorter than their co-isogenic Berlin flies (Supplementary Fig. 1e), corroborating that deletion of this genomic region reduced lifespan and was not due to a bias in the genetic background. Finally, to check if the deletion of the genomic region encompassing snoRNA:Ψ28S-1153 presents a sexual dimorphism, we also determine longevity

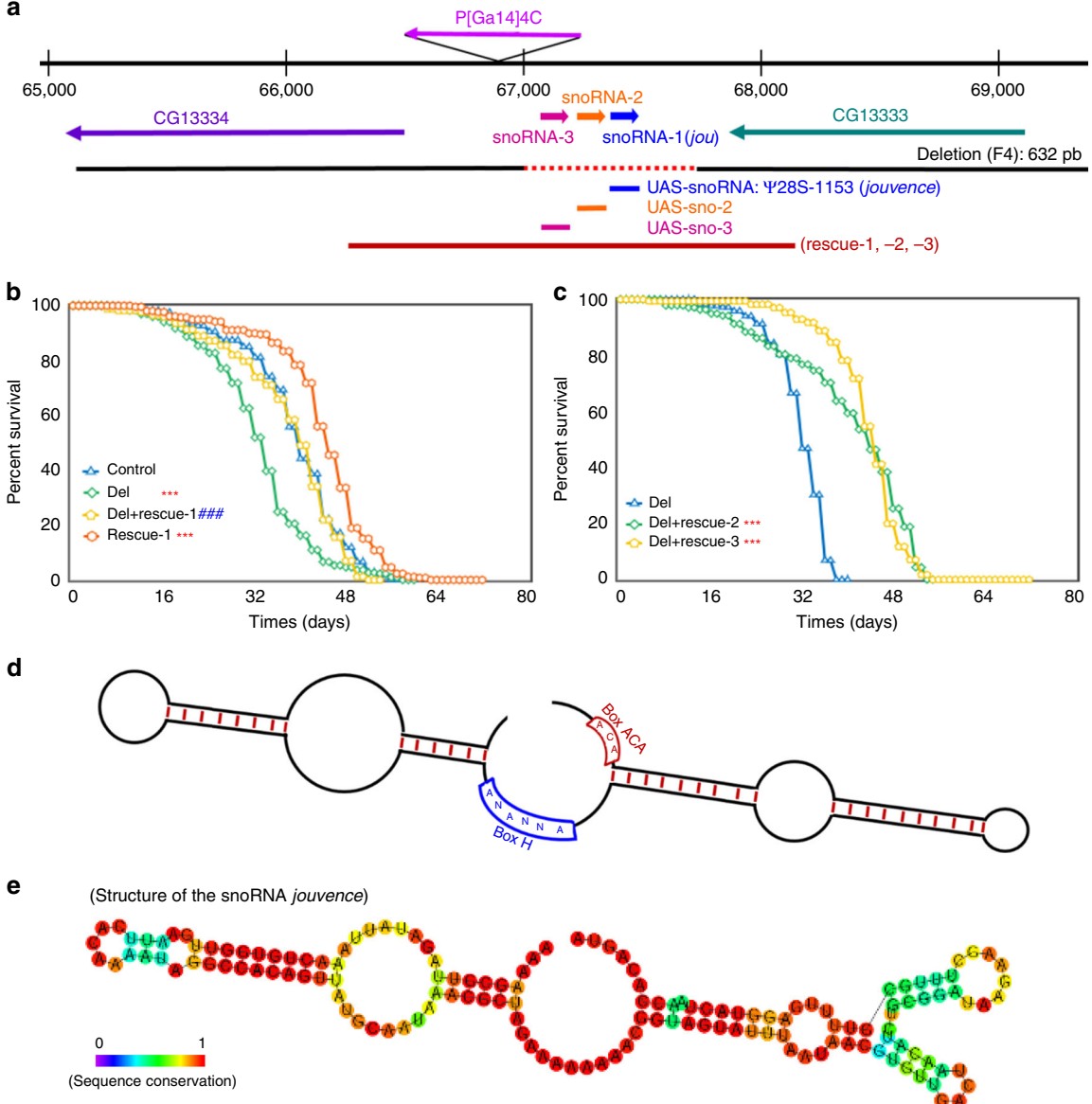

**Fig. 1 Molecular map of the P[Gal4]4C locus and features of the snoRNA:Ψ28S-1153. a** Genomic map of the P[Gal4]4C locus. The snoRNA:Ψ28S-1153 (*jouence*), as well as the two other putative upstream snoRNAs (sno-2 = snoRNA:2R:9445205 and sno-3 = snoRNA:2R:9445410), are in the inverse orientation of the two encoding genes: CG13333 and CG13334. The deletion (F4) of 632 bp (red dotted line) includes the three putative snoRNAs. The 1723 bp genomic DNA fragment (genomic-rescued named: rescue-1, rescue-2, and rescue-3) (red line) used to generate transgenic flies. The snoRNA-*jou* fragment (blue bar) of 148 bp used to generate the UAS-jou[8M] and pJFRC-MUH-jou[1M] constructs, the 166 bp fragment (orange bar) corresponding to the sno-2, and the 157 bp fragment (pink bar) corresponding to the sno-3. **b** Decreasing cumulative of Control (CS), Del (deletion F4), Del + rescue-1 and rescue-1. Note: the lifespan determination of these 4 groups of flies (genotypes) have been performed simultaneously, in parallel (for number of flies, age in days at % mortality, and detailed Statistics, see Supplementary Information, Table 1) (*** = $p < 0.001$, compared to Control; ### = $p < 0.001$ compared to Deletion). **c** Similarly to (**b**), decreasing cumulative of females Del (deletion F4), Del+rescue-2, Del+rescue-3. In Del+rescue-2, Del+rescue-3, the lifespan is lengthened (for detailed Statistics: see Supplementary Information, Table 1). **b**, **c** *p*-value calculated by log-rank test. **d** Schematic representation of a H/ACA snoRNAs structure. **e** Schematic representation of the snoRNA:Ψ28S-1153 (*jouence*), harbouring a typical H/ACA box structure. For **b**, **c** Source data are provided as a Source Data file.

in males (Supplementary Fig. 1f). Males deleted flies live longer than control WT. Interestingly, differences in lifespan between the sexes have already been reported in many species related to various longevity genes and/or signalling pathways, as for example, the IIs[38–40]. However, for the sake of simplicity, and since the precise determination of the sex difference in lifespan is a complex and full topic per se, here as a first step, we restrict our study to females.

**The snoRNA:Ψ28S-1153 is expressed in the gut epithelium.** Since only the snoRNA:Ψ28S-1153 has been identified in the bioinformatic study as a real/canonical snoRNA[34], we hypothesize that this snoRNA might be potentially the only true functional snoRNA in this locus. In order to determine in which tissues snoRNA:Ψ28S-1153 could be expressed and required to affect lifespan, we performed ISH on the whole flies using fluorescently-labelled probes (Fig. 2). We then performed ISH

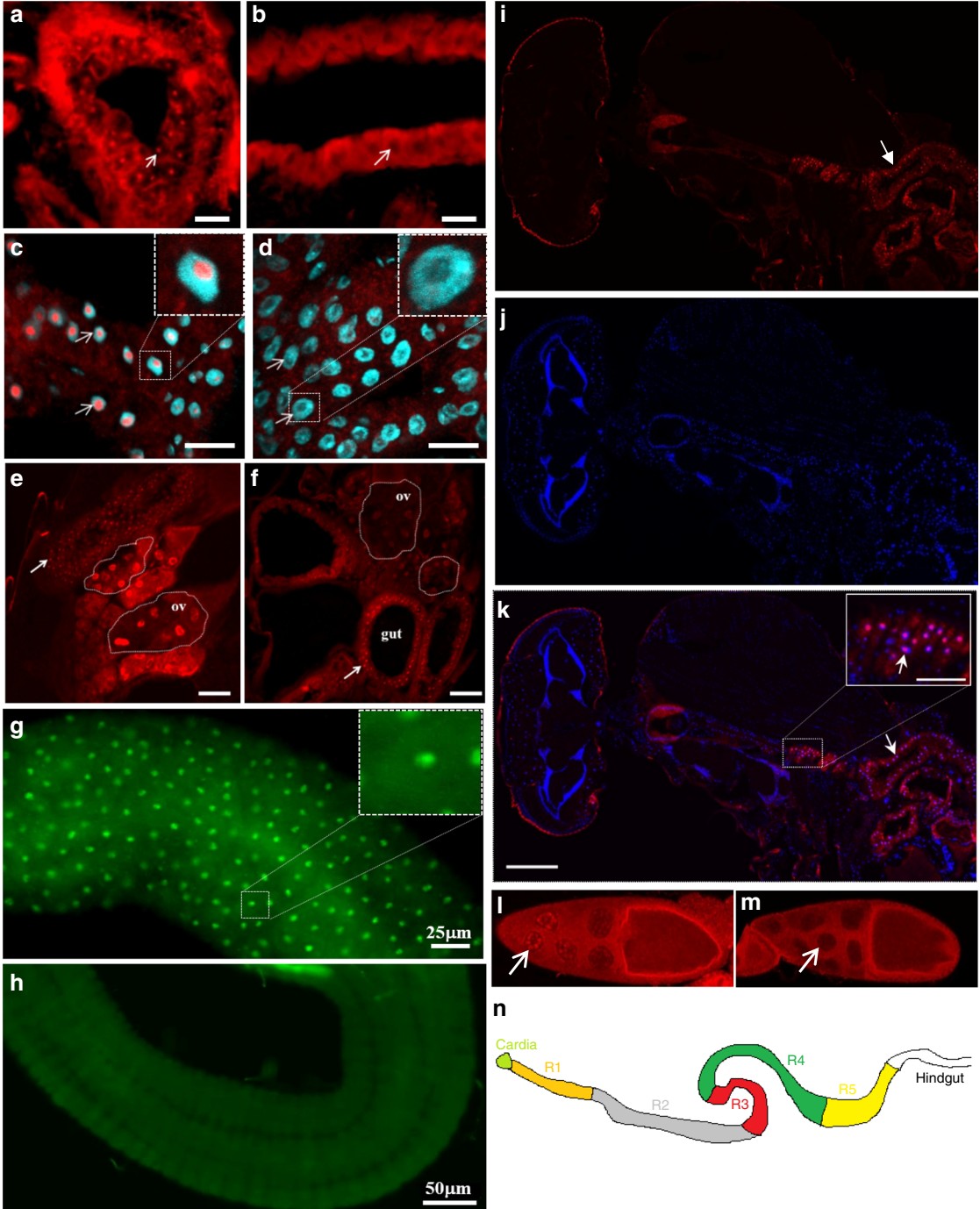

**Fig. 2 snoRNA:Ψ28S-1153 (*jou*) is expressed in the epithelium of the gut.** In situ Hybridization (ISH) of the snoRNA-*jou* on whole fly (Cryostat section) reveals a restricted expression in the epithelium of the gut (red dots: white arrow) in Control (**a**), but absent in the deletion (**b**) (white arrow). **c** Magnified view taken with a confocal (×63) revealed that *jou* is expressed in the nucleolus (DAPI counter-staining), but absent in deletion (**d**) (scale bar = 25 μm). Inset (magnification) showing that *jou* is located in the nucleolus. **e**, **f** ISH (Cryostat section) of *jou* on transgenic whole flies (rescue-1 and Del+rescue-1). In rescue-1 (Wild-Type background) (**e**), *jou* expression was seen in the gut epithelium (red dots: white arrow) and ovaries (ov) (see also Supplementary Fig. 2), while expression was restricted to the gut epithelium in Del+rescue-1 and absent in the ovaries (ov) (**f**) (scale bar = 50 μm). **g**, **h** ISH, revealed with a FITC-labelled tyramide (green), on dissected gut of Wild-Type CS flies, in the midgut (**g**), *jou* is detectable in each cells of the epithelium. Inset (magnification): *jou* is visible in the nucleolus. **h** In the hindgut, *jou* is not expressed. **i–k** ISH on a whole Wild-Type CS fly (Cryostat section). **i** *jou* expression revealed with tyramide-Cy3 (white arrow). **j** DAPI-staining labelling the nucleus. **k** Overlay of (**i**), **j** Showing the expression of *jou* only in the epithelium of the gut (white arrow), but no expression in the nervous system. Notes: (1) ovaries are also positive but they are not visible on this section, (2) the red colour surrounding the head in (**i**) and (**k**) corresponds to the auto-fluorescence of the retina. Insert in (**k**) magnification view. red dots = *jou* in the nucleolus of the epithelium of the gut (scale bar = 200 μm, Insert: scale bar = 50 μm). **l**, **m** ISH on ovaries. Stacks of confocal images showing *jou* expression in the nucleolus of Control (**l**) and absent in deletion (**m**). **n** Schematic drawing of the gut showing its different main regions.

with this probe and found that it was expressed in gut epithelial nucleoli and the proventriculus (Fig. 2a, c), but as expected, its expression was absent in deletion line (Fig. 2b d). Moreover, a counter-staining with DAPI suggest that it is localised in the nucleolus (Fig. 2c and d). In addition, whole-body examination revealed that the snoRNA:Ψ28S-1153 was also expressed in ovarian nurse cell nucleoli in control flies (Fig. 2e, f and l, m and Supplementary Fig. 2), but not in any other tissues, including the nervous system (see Fig. 2i, j, k for the expression pattern of the whole fly, including the head, the nervous system and the thorax). Similarly, when the rescue construct was combined with the deletion line, snoRNA:Ψ28S-1153 expression was restricted to gut epithelial cells, though surprisingly, not in the ovaries (Fig. 2e, f). Additional ISH performed on dissected gut of Wild-Type flies confirmed that the snoRNA:Ψ28S-1153 was expressed more precisely in the midgut (Fig. 2g), but not in the hindgut (Fig. 2 h) (see Fig. 2n for a schematic view of the gut). These results suggested that effects on longevity of snoRNA:Ψ28S-1153 were likely due to its expression in the gut epithelial cells, and not in the ovaries, although at this stage of this study, the role of the two other putative snoRNAs could not be completed excluded.

The gut epithelium is composed of four cell-types: enterocytes (ECs), enteroblasts (EBs), entero-endocrine cells (EEs) and intestinal stems cells (ISCs)[41–44]. We specifically marked these cell types by combing ISH with GFP expression driven by cell type-specific drivers. To target and label the ECs, we used the Myo1A-Gal4 driver (NP1-Gal4), an enhancer trap in myosin 1A gene, encoding a gut specific brush border[45]. As described by Jiang et al.[46], Myo1A-Gal4 is strongly expressed in all midgut ECs, while no expression was detected in ISCs, EBs, EEs, or visceral muscle. To label the EBs, we used the Su(H)GBE-Gal4[44], while the ISCs were marked by both esg-Gal4 and Dl-Gal4. Amongst these, only EC-specific expression (Myo1A-Gal4 showed *jou* co-labelling with the GFP reporter (Supplementary Fig. 3), indicating that its localisation was restricted to enterocytes.

**The snoRNA:Ψ28S-1153 in adulthood extends lifespan**. To corroborate the previous results obtained with the genomic res-cued transgene (Del+rescue-1) and confirm that the longevity effect is due to the snoRNA-Ψ28S-1153 in the epithelial cells of the gut, and not to the two other putative snoRNAs, we generated a p[UAST-snoRNA-Ψ28S-1153] (named pUAS-jou[8M]) transgenic line. Then, we targeted this unique snoRNA expression specifically to enterocytes in deletion background using Myo1A-Gal4 line (Del,Myo1A>UAS-jou[8M]) and verified by ISH that the snoRNA was indeed correctly expressed in the enterocytes (Supplementary Fig. 3e). Furthermore, the expression of the snoRNA-Ψ28S-1153 specifically in enterocytes was sufficient to increase the longevity of the flies (Fig. 3a), compared to their two respective controls (Gal4 line alone in heterozygous and UAS-jou[8M] alone in het-erozygous) (for a second replicate, see Supplementary Fig. 4a). Remark that in the control line bearing the P[Gal4] driver in heterozygous (Del,Myo-Gal4/Del) the longevity is increased compared to the *jou*-deleted flies (Fig. 1b). This increase is due to an insertional effect of the P[Gal4] driver itself, since the recovery (de-recombination) of the deletion from this line, de novo gives a similar longevity to the original deletion line (Supplementary Fig. 4f). In addition, to investigate if the two other putative snoRNAs (snoRNA:2R:9445205 and snoRNA:2R:9445410, short-named sno-2 and sno-3 respectively) localized just upstream to the snoRNA:Ψ28S-1153 were also involved in the lifespan determi-nation, the targeted expression of sno-2 and sno-3 in ECs (Del, Myo1A>UAS-sno-2 and Del,Myo1A>UAS-sno-3) (Fig. 3b, c), did not increase lifespan of the deletion line encompassing snoRNA:Ψ28S-1153, indicating that only the snoRNA:Ψ28S-1153

expression in ECs is capable to rescue lifespan among these three snoRNAs. To further support this demonstration, we also observe a rescue of longevity defects compared to their co-isogenic control flies by expression of snoRNA-Ψ28S-1153 (UAS-jou[8M]) with a second Gal4 driver line, Mex-Gal4[47], which is also expressed in the epithelium of the gut though not exclusively in the ECs[48] (Fig. 3d). In the deletion background, the targeted expression in the epithelial cells of the gut by Mex-Gal4 also increases lifespan. In addition, to exclude that putative insertional effect of the UAS-jou[8M] could lead to some longevity effects per se, we have gen-erated a second and independent UAS-jou construct into the pJFRC-MUH vector (a more recent and updated pUAST vector, which also contains directed insertional sites)[49] and generate a transgenic fly line (named pJFRC-MUH-jou[1M]) inserted at the attP2 site located at position 68A4 on the third chromosome. We then repeated the longevity experiments using the same P[Gal4] driver lines (Del,Myo1A-Gal4>pJFRC-MUH-jou[1M] and Del,Mex-Gal4>pJFRC-MUH-jou[1M]). Both of them lead to an increase of longevity (see Supplementary Fig. 4b, c), a similar result obtained with the UAS-jou[8M] construct, demonstrating that it due to the expression of the snoRNA-*jouvence*.

We then wanted to exclude that expression of snoRNA:Ψ28S-1153 by Myo1A-Gal4 and the Mex-Gal4 during development rescues longevity defects. We, therefore, generated a conditional Gal4 Gene-Switch line (Mex-GS) to allow expression of the UAS-jou[8M] only in adult stages. As a first step, we confirm that the Gene-Switch construct and the transgenic flies were functional and the induction of the reporter transgene occurs following feeding of the flies with the RU486. Supplementary Fig. 5f, g (and see Supplementary Fig. 14 for the whole animal) shows that in RU486 fed flies of Mex-GS>UAS-GFP, as expected, the expres-sion of the GFP is easily detectable in the gut of adult flies, validating the system. Second, in the deletion genetic background, feeding flies with RU486 starting just after the hatching and so, during adulthood, led to an increase of lifespan compared to the sibling control flies without RU486 (Fig. 3e). Similar results were also obtained with the pJFRC-MUH-jou[1M] construct (Del,Mex-GS>pJFRC-MUH-jou[1M]) (Supplementary Fig. 4d). In comple-ment, to verify and rule-out that the positive longevity effect is not due the RU486 itself, we determine the longevity in control flies without the Mex-GS driver (Del;pJFRC-MUH-jou[1M]) (Sup-plementary Fig. 4e; see also other RU486 controls in Supplemen-tary Fig. 5c–e). Interestingly, we do not observe any difference in longevity between the two groups of same genotype flies: no-fed versus fed with RU486. In contrast, similar RU486 inducible experiments targeting either the UAS-sno-2 or the UAS-sno-3 did not yield to an increase of lifespan but rather to a decrease of lifespan (Fig. 3f, g) indicating that the targeted expression of these two snoRNAs respectively could be deleterious (a result similar to the one's obtained with the Myo1A-Gal4 driver: Fig. 3b, c), and consequently confirming that *jouvence* is the snoRNA responsible for the lengthening of lifespan.

To confirm the snoRNA:Ψ28S-1153 expression levels and to temptingly correlate it to longevity, we performed a quantitative PCR on the dissected gut (Supplementary Fig. 6a–c). As expected, the snoRNA:Ψ28S-1153 is not detected in deletion line, whereas its expression is rescued in genomic-rescued line (Del+rescue-1) compared to Control (CS). Moreover, its expression is signifi-cantly increased (4 folds) in rescue-1 transgenic flies (in WT genetic background), confirming overexpression in this line that carries four copies of the snoRNA-*jouvence*. We also quantify the level of the snoRNA on dissected gut (Supplementary Fig. 6b) in the two gut-driver lines. The levels of the snoRNA:Ψ28S-1153 was increased about 500-fold for the Myo1A and about 600-fold with Mex-Gal4 (driving UAS-jou[8M]). Similarly the Mex-Gene-Switch

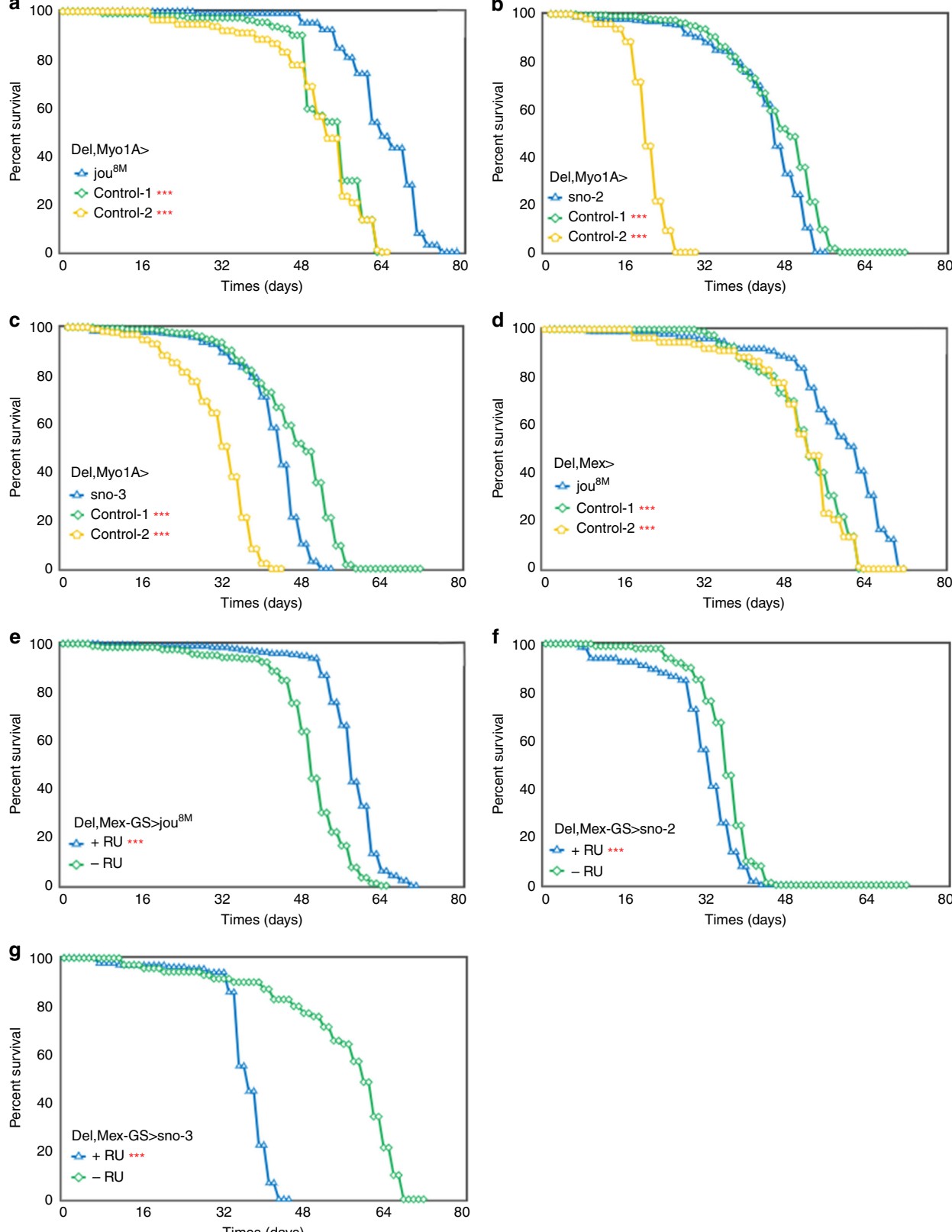

driving the snoRNA-*jou* (Del,Mex-GS>UAS-jou$^{8M}$) induced an increase of *jouvence* after RU486 induction (up to 5 folds) (Supplementary Fig. 6c). Thus, increased levels of snoRNA:Ψ28S-1153 were always correlated with increased longevity.

**Overexpression of *jouvence* in WT flies increases lifespan.** Since in the deletion background, the targeted expression of the snoRNA:Ψ28S-1153 specifically in the enterocytes rescued the phenotype, we wondered if the overexpression of this snoRNA in

**Fig. 3 Targeted expression of _jou_ in enterocytes is sufficient to rescue the longevity.** Longevity test results (survival curve—decreasing cumulative) of the targeted expression of the snoRNA:Ψ28S-1153 specifically in the enterocytes in deletion compared to their controls. **a** Del,Myo1A-Gal4>UAS-jou8M expression in enterocytes is sufficient to increase lifespan. (Control-1 = Del,Myo1A-Gal4/Del, Control-2 = Del/Del;UAS-jou8M /+). **b, c** The expression of the two other snoRNAs: sno-2 (**b**), or the sno-3 (**c**), do not lead to an increase of lifespan, and even it seems to be slightly deleterious. **d** A second gut driver line, Mex-Gal4 (Del,Mex-Gal4>UAS-jou8M) which also targets the expression in enterocytes is sufficient to increase lifespan. **e** The conditional Mex-GS flies (Del,Mex-GS>UAS-jou8M) fed with the RU486 only in adulthood, which triggers the expression of the snoRNA-jou, is sufficient to increase lifespan compared to the non-fed (non-induced) sibling flies. **f, g** Mex-GS flies fed with RU486 only in adulthood, which triggers the expression of the sno-2 (**f**) or sno-3 (**g**), do not increase lifespan compared to the non-fed (non-induced) sibling flies, but it is rather deleterious. In each longevity panel, Control-1 = Del,Gal4-driver/Del, meaning the Gal4 driver without the UAS-construct), Control-2 = Del/Del;UAS-construct/+, meaning the UAS-jou8M, or sno-2, or sno-3, in heterozygous, without the Gal4 driver (Notice: same nomenclature for all other rescued longevity Figures). (*** = $p < 0.001$) (for number of flies, age in days at % mortality, and detailed statistics, see Supplementary Information, Table 1). _p_-value calculated by log-rank test. For all panels, Source data are provided as a Source Data file.

a Wild-Type genetic background (CS) could be beneficial leading to an increase of lifespan or in contrast, be deleterious. We used the same driver lines to overexpress the snoRNA:Ψ28S-1153 up to 13-fold (Myo1A-Gal4) and 28-fold (Mex-Gal4) over that of endogenous expression (Fig. 4f). Both Gal4-drivers expressing the snoRNA gave similar results, showing increased longevity (Fig. 4a, b), compared to their co-isogenic controls flies. Again here, similar results were obtained with the pJFRC-MUH-jou1M construct (Mex-Gal>pJFRC-MUH-jou1M) (Supplementary Fig. 5a). Moreover, conditional adult-specific expression via RU486 feeding in the Mex-GS line (30X increased expression: Fig. 4g) also increased lifespan compared to the non-induced flies (Fig. 4c), as well as with the second independent transgenic line (Mex-GS>pJFRC-MUH-jou1M) (Supplementary Fig. 5b). Again here, to ensure that the longevity effect is not a secondary effect due to the RU486 itself, three independent controls have been conducted: the Gene-Switch driver alone without the UAS-jouvence (Mex-GS/CS), and the two independent UAS-jouvence lines without the driver line (UAS-jou8M/CS and pJFRC-MUH-jou1M/CS): all three controls don't show any difference in longevity comparing fed versus non-fed flies with RU486 (Supplementary Fig. 5c–e). Finally, the induction of the overexpression in adulthood, by the Mex-GS, of the sno-2 or the sno-3 did not alter lifespan compared to their respective co-isogenic control flies (Fig. 4d, e), corroborating that the snoRNA-jou is critical for lifespan determination. In conclusion, since the snoRNA:Ψ28S-1153 increases lifespan when it is overexpressed in the epithelium of the gut, we conclude that it could be considered as a longevity gene, and therefore we named it "_jouvence_" (_jou_) (which means "youth" in English).

**Overexpression of _jouvence_ prevents the gut hyperplasia.** In aged flies, the intestinal epithelium degenerates leading to intestinal hyperplasia[50,51]. Then, we wonder if the deletion of the snoRNA-_jouvence_ affects the homoeostasis of the gut and leads to cellular lesions. To assess such hyperplasia, we count the number of enterocytes both in young (7 day-old) and old (40-day-old) flies. We observe that this number is increased in the _jou_-deleted flies compared to their respective WT controls (Fig. 5a), indicating a hyperplasia. Interestingly, such increase in the number of enterocytes is not observed in the old (40-day-old) flies carrying the genomic-rescued transgene flies (Del;rescue-1), as well as in old flies carrying a gut targeted expression of _jouvence_ compared to their respective control flies (Del,Mex-Gal4>jou1M and Del, Myo1A-Gal4>jou1M). In addition, since the expression of _jouvence_ (in the deletion background) in the gut is sufficient to rescue and even prevent the hyperplasia, we wonder if the overexpression of _jouvence_ in a Wild-Type background could be sufficient to prevent such hyperplasia in WT flies. Figure 5a shows that indeed, such overexpression is sufficient, although to less extent, to prevent the hyperplasia (Mex-Gal4>jou1M and Myo1A-Gal4>jou1M).

As the gut is a high-turnover tissue, the number of enterocytes might reflect the proliferative homeostasis of the gut all along the life. To evaluate the origin of this hyperplasia, we quantify the number and the proliferation of the intestinal stem cells (ISCs) as well as the putative mis-differentiation of ISCs daughter cells (enteroblasts: EBs), using different labelling approaches. First, we use the Delta-Gal4 to drive the pUAS-GFP-nls to count the number of ISCs[52]. Figure 5b shows that the number of Delta labelled cells is increased in _jou_-deleted flies. Second, to assess the proliferation of the ISCs, we count the number of anti-PH3 positive cells[46,51]. Interestingly, it reveals (Fig. 5c) a quite similar pattern to the Fig. 5b, indicating that the division of cells follows quite faithfully the number of Dl-labelled ISCs. In a step further, to alternatively and independently assess the division of the cells, we use the EdU labelling[53] (a recent updated alternative to BrdU) which labels the dividing as well as the endoreplicative cells. In Fig. 5d we can see that in young flies, the number of EdU labelled cells is increased in _jou_-deleted flies (in accordance to the higher number of endoreplicative ECs), while in aged flies, the number of EdU cells is increased in WT but not statistically different in _jou_-deleted flies, which seems also to reflect quite faithfully the number of Dl positive cells (Fig. 5b). Altogether, these result suggests that the higher number of ECs observed in _jou_-deleted flies result from a higher proliferation of the cells as revealed by the number of Dl, anti-PH3 and EdU positive cells.

**_jouvence_ modulates the endoreplication in enterocytes.** In _Drosophila_, several tissues, as gut, fat body, salivary glands, Malpighian tubules, trachea, muscle and epidermis contain polyploïd cells due to endoreplication cycle[54,55]. The ISH has revealed that the snoRNA-_jouvence_ is specifically expressed in two endoreplicative cell types, the enterocytes and ovarian nurse-cells, but not in other endoreplicative tissues. Thus, we wonder if the endoreplication is affected in _jou_-deleted flies. We then assess, by two independent and commonly used approaches, the endoreplication in the gut. First, we quantify the amount of DAPI-staining fluorescence of the nucleus, which reflect the quantity of DNA[56]. Figure 5e reveals that the quantity of DNA is reduced in _jou_-deleted flies compared to WT-Controls, indicating less endoreplication. Interestingly, the quantity of DNA (endoreplication) is rescued in flies bearing the genomic-rescued transgene (Del+rescue-1), as well as in deletion flies with a targeted expression of _jouvence_ specifically in the gut (Del,Mex-Gal4>UAS-jou8M). In a step further, we also quantify the endoreplication in flies overexpressing the snoRNA-_jouvence_ in WT genetic background, and more particularly when expressed only in adulthood using the Gene-Switch system. After feeding RU486 to induce the _jouvence_ expression, the endoreplication is increased compared to the non-fed RU486 control flies (Mex-GS>jou1M), indicating that _jouvence_ is involved in the control of the polyploidy in ECs. This last result also demonstrates that _jouvence_, among the three snoRNAs of the locus, is

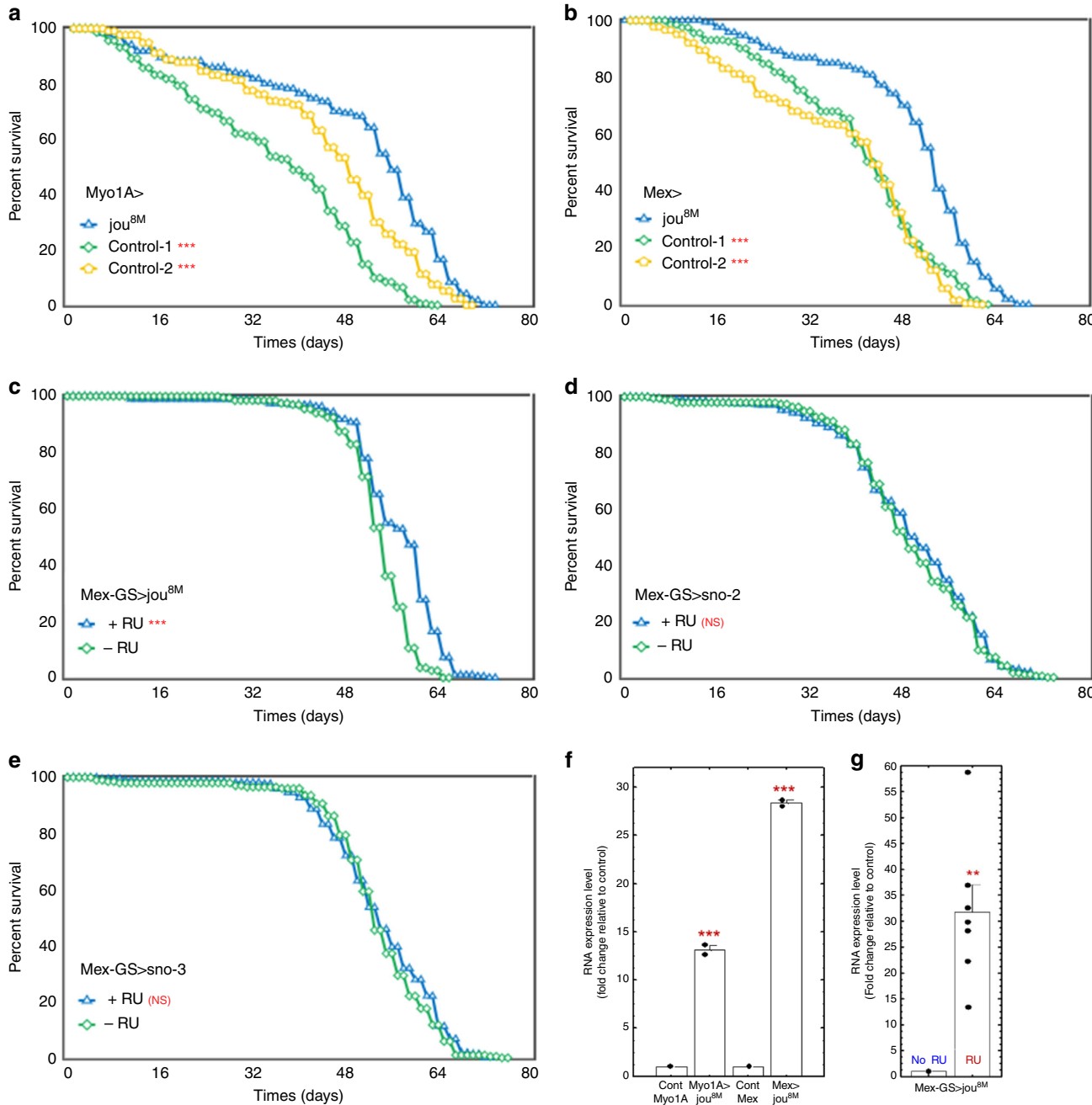

**Fig. 4 Overexpression of *jou* in the enterocytes increases lifespan.** Longevity test results (survival curve − decreasing cumulative) of the targeted expression of *jou* specifically in the enterocytes in Wild-Type genetic background compared to their controls. Myo1A-Gal4>UAS-jou[8M] (**a**) and Mex-Gal4>UAS-jou[8M] (**b**) overexpression in enterocytes is sufficient to increase lifespan. In each longevity panel, Control-1 = Gal4-driver/+, meaning the Gal4 driver without the UAS-construct, while Control-2 = UAS-construct/+, meaning the UAS-jou[8M], or sno-2, or sno-3, without the Gal4 driver (Notice: same nomenclature for all other overexpression longevity Figures). **c** Overexpression of *jou* only in adulthood is sufficient to increase lifespan (Mex-GS>UAS-jou[8M] flies fed with RU486 only in adulthood). **d, e** Overexpression of the sno-2 (**d**) or sno-3 (**e**) only in adulthood do not have any effect on lifespan (Mex-GS>UAS-sno-2 or sno-3 flies fed with RU486 only in adulthood) (*** = $p < 0,001$) (for number of flies, age in days at % mortality, and detailed statistics, see Supplementary Information, Table 1). *p*-value calculated by log-rank test. **f** RT-qPCR (Taqman) results of the targeted overexpression of *jou* specifically in the enterocytes compared to their controls Myo1A-Gal4, and Mex-Gal4, respectively. Two assay-repetitions were done (*n* = 2) for each genotype. **g** RT-qPCR (Taqman) results of the RU486 induced targeted overexpression of *jou* specifically in the enterocytes only in adulthood in Mex-GS (*n* = 4), compared to the non-induced controls flies (*n* = 7). For **f, g** the snoRNA-*jouvence* is normalized to the level of rp49. For each condition, 30 guts were dissected for the total RNA extraction. Statistics: compared to control Myo/CS, or Mex/CS, or to the non-induced Mex-GS: (*p*-values) (*$p < 0,05$; **$p < 0,005$; ***$p < 0,0005$). Errors bars represent the mean ± S.E.M. (*p*-value were calculated using the one-way ANOVA followed by a TUKEY test). For all panels, Source data are provided as a Source Data file.

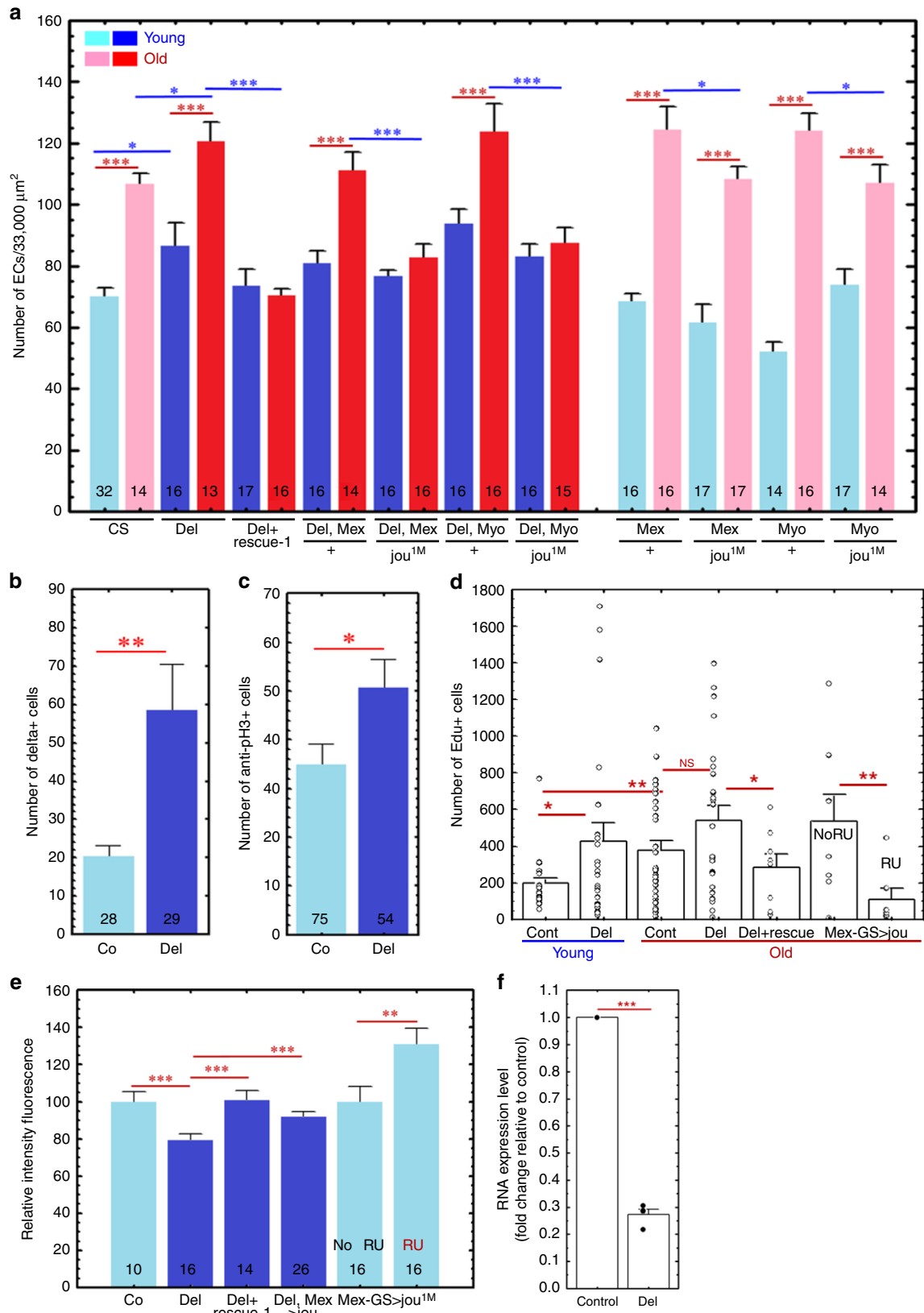

sufficient by itself to modulate the endoreplication. Second, to reinforce such measurement, we also quantify, by RT-qPCR, the Cyclin-E, one of the main genes known to be involved in the endoreplicative cycle[53,56,57] (Fig. 5f). In accordance to the decrease of the amount of DNA quantified by the DAPI-staining,

Cyclin-E RNA expression is decreased in *jou*-deleted flies, a result also supported by the RNA-Seq analysis (see Supplementary Data 2, line 147). Therefore, altogether, these results indicate that the snoRNA *jouence* is involved in the endoreplication in ECs.

**Fig. 5 Deletion of *jouvence* increases the hyperplasia in old flies. a** Number of enterocytes (ECs) in young (7 day-old: light and dark blue) and aged (40-day-old: light and dark red) flies (light = in WT and dark = in deletion). As expected, the number of ECs is increased (hyperplasia) in aged WT flies compared to young (7 day-old) flies. However, this number is more increased in *jou*-deleted flies (dark blue and dark red compared to light blue and light red). Remarkably, the number of ECs is not increased in old genomic-rescued flies (Del;rescue-1), as well as in targeted expression of *jou* in the gut (Del, Myo1A-Gal4>UAS-jou1M and Del,Mex-Gal4>UAS-jou1M). Finally, the overexpression of *jou* in WT genetic background prevents the hyperplasia (Myo1A-Gal4>UAS-jou1M and Mex-Gal4>UAS-jou1M). **b** The number of Dl+ cells (Dl-Gal4>UAS-GFP-nls) in WT control and *jou*-deleted flies. **c** The number of anti-PH3 cells. **d** The number of EdU labelled cells, both in young (blue) and old flies (red). Co-young: n = 22, Del-young: n = 24, Co-old: n = 31, Del-old: n = 26, Del+rescue-1-old: n = 8, Mex-GS>jou, no RU: n = 8, with RU: n = 7). **e** Relative Intensity of DAPI-fluorescence used to evaluate the endoreplication. **a–e** The number at the bottom of each graph represents the number of animals assayed (n). **f** RT-qPCR (SybrGreen) performed on the total RNA extracted from the gut on the cyclin-E gene, used to evaluate/quantify the endoreplication. The expression of the cyclin-E is importantly reduced in *jou*-deleted flies (n = 4 for each genotype). Statistics: *$p < 0,05$; **$p < 0,005$; ***$p < 0,0005$. Errors bars represent the mean ± S.E.M. (p-value were calculated using the one-way ANOVA followed by a TUKEY test). For all panels, Source data are provided as a Source Data file.

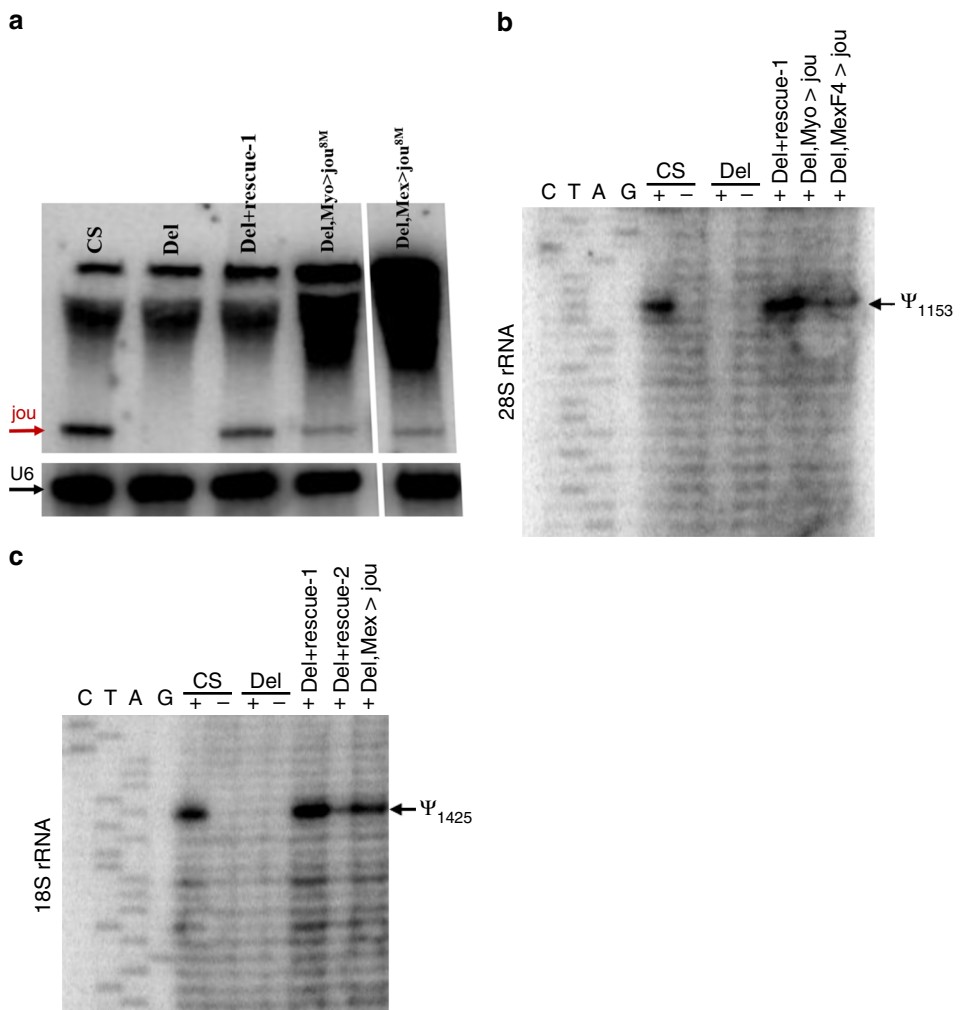

**Fig. 6 *Jouvence* is correctly processed and performs the pseudouridylation in vivo. a** Northern blot reveals that jouvence is correctly processed at approximately 148 nucleotides in WT control flies, missing in deletion line, and rescued in genomic-rescued flies (Del;rescue-1) as well as in gut targeted *jou* (Myo1A-Gal4>jou8M and Mex-Gal4>jou8M). U6 used as loading control. **b** The snoRNA-*jouvence* pseudouridylates the 28S-rRNA at position 1153 in WT controls, while it is missing in deletion, and rescued in genomic-rescued flies (Del;rescued-1) as well as in gut targeted *jou* (Myo1A-Gal4>jou8M and Mex-Gal4>jou8M). **c** idem for the 18S-rRNA, except that the pseudouridylation occurs at the position 1425. For all panels, Source data are provided as a Source Data file.

**Jouvence is correctly processed in vivo (northern blot).** Though we have shown by RT-PCR that *jouvence* is expressed in the gut, to provide a more direct measurement of the RNA size and ensures it is correctly processed, we performed northern blots. In Fig. 6, we can observe a band at ~150 nucleotides corresponding to the expected size of *jouvence*, both in control WT as well as in flies carrying either the genomic-rescued transgene (Del+rescue-1) or

in flies with specific targeted expression of *jouvence* in the gut (Del, Myo1A-Gal4>jou8M and Del,Mex-Gal4>jou8M).

**The snoRNA-*jouvence* pseudouridylates the rRNA.** According to the reported bioinformatic study in which the snoRNA:Ψ28S-1153 has been primarily annotated[34], the snoRNA-*jouvence*

(snoRNA:Ψ28S-1153) is predicted to catalyse the modification of a 28S-rRNA base, at position 1153. Based on the tissue-specific expression of *jou*, and to grasp insights into the molecular mechanism of action of *jou*, we check if the rRNA, extracted from the gut, are indeed pseudouridylated. Figure 6b shows, as expected, that the 28S-rRNA is pseudouridylated at position 1153 in WT-Control flies, whereas it is not pseudouridylated in *jou*-deleted flies, indicating that *jou* is required for this site specific pseudouridylation. Moreover, the re-expression of *jouvence* in *jou*-deleted flies carrying the genomic-rescued transgene (Del+rescue-1) as well as in flies with specific targeted expression of *jouvence* in the gut (Del,Myo1A-Gal4>jou$^{8M}$ and Del,Mex-Gal4>jou$^{8M}$) demonstrates that the pseudouridylation mechanism is rescued. Additionally, we also show that *jouvence* pseudouridylates the 18S-rRNA at position 1425 in WT control flies (Fig. 6c), which is absent in *jou*-deleted flies, while it is rescued in genomic-rescued flies (Del+rescue-1 and Del+rescue-2) and gut specific targeted expression (Del,Mex-Gal4>jou$^{8M}$).

**Several genes are deregulated in *jou*-deleted flies**. In a step further, to grasp additional insights into the molecular mechanism of action of *jou*, and to better understand *jou* function in longevity we performed a transcriptomic analysis (RNA-Sequencing) of the gut tissue comparing Wild-Type to *jou*-deleted flies. Based on a standard stringency of 2-fold changes, RNA-Seq reveals that 314 genes were upregulated, while 319 genes were downregulated (Fig. 7a, b and Supplementary Data 1, 2). Moreover, a bioinformatic analysis based on KEGG (Kyoto Encyclopedia of Genes and Genomes)[58], has revealed that 20 pathways are statistically upregulated (Fig. 7c and Supplementary Data 3), the most one being the metabolic pathway, encompassing 55 deregulated genes, while only 6 pathways are downregulated (Fig. 7d and Supplementary Data 4), the most down-deregulated one being the Glutathione metabolic pathway (although this deregulation is less severe). In addition, a Gene Ontology (GO) analysis revealed that the majority of the deregulated genes have a catalytic activity (Supplementary Fig. 7a, b and Supplementary Data 5, 6).

To confirm some of the results obtained by the RNA-Seq analysis, we selected some of the most up- or downregulated genes to perform RT-qPCR on gut tissue. Figure 8 shows that *GstE5* (*Glutathione S transferase E5*), *Gba1a* (*Glucocerebrosidase 1a*), *LysB* (*Lysozyme B*), and *ninaD* (neither inactivation nor afterpotential D) are strongly upregulated in deletion line, whereas CG6296 (lipase and phosphatidylcholine 1-acylhydrolase predicted activities), and *Cyp4p2* (*Cytochrome P450-4p2*) are strongly downregulated, confirming the results of the RNA-Seq. The regulation of these genes by *jou* was confirmed by the analysis of the rescued genotypes. For *GstE5* gene, the level of RNA is rescued in genomic-rescued line (Del+rescue-1) as well as in Myo-Gal4 expressing *jou* in deletion background (Del,Myo1A>UAS-jou$^{8M}$) specifically in enterocytes. For *Gba1a* and *LysB*, the rescue is quite similar to *GstE5*. However, for the *ninaD*, the regulation seems to be more complex, since the level of RNA is inverted (importantly increased) in genomic-rescued line (Del;rescue-1) and not rescued in Del, Myo1A>UAS-jou$^{8M}$ line, suggesting that for this gene the fine regulation of the RNA level is more complex and might putatively involve other factors. For the two selected downregulated genes (CG6296 and *Cyp4p2*), similar to *ninaD*, the regulation seems also to be complex since none of them are rescued by the genomic-rescue line (Del,rescue-1) neither by the Myo1A-Gal4 (Del, Myo1A>UAS-jou$^{8M}$), although it is partially rescued (statistically different) in Del,rescue-1 and Del,Myo1A>UAS-jou$^{8M}$ for CG6296. These results suggest that the longevity function of the snoRNA-*jou* relies on its ability to regulate either the mRNA level or the ribosomal RNA stability, or even perhaps the gene expression in the

enterocytes. However, such precise mechanisms remain to be further studied.

**Overexpression of *jou* deregulates only few genes**. RNA-Seq analysis has revealed that few hundreds of genes are either up- or down-regulated in *jou*-deleted flies, which leads to a shorten lifespan. Inversely, we wonder which genes are modified in flies overexpressing *jou* in a WT genetic background, whose yields to an increase of lifespan. Thus, we performed a second RNA-Seq on the dissected gut of flies expressing conditionally *jouvence* only in adulthood using the Gene-Switch system (Mex-GS>jou$^{8M}$), comparing RU486-fed flies (induced expression) with no-RU486 fed flies (control-non-induced). Surprisingly, it reveals that only 9 genes are deregulated in *jouvence* overexpressing flies, based on a standard stringency of 2-fold changes (Fig. 9 and Supplementary Data 7). The analysis of pathways enrichment (KEGG analysis)[58] reveals that they are involved principally in the spliceosome and the longevity regulating pathway (Supplementary Data 8), the most upregulated one being the Hsp70Bb gene (Heat-shock-protein-70Bb) (Supplementary Fig. 7c for the GO analysis and Supplementary Data 9).

**Restoring mRNA levels of *ninaD* or CG6296 rescues lifespan**. In a step further, to gain downstream mechanistic insights on how the snoRNA *jouvence* transmits its effect on longevity, we use a genetic approach. First, we select one of the most upregulated genes revealed by the RNA-Seq (*ninaD*), to direct a RNAi, in the aim to decrease and restore its mRNA level. We show that the targeted expression of a *ninaD*-RNAi specifically in the gut, and more precisely only in adulthood (Del,Mex-GS>UAS-ninaD-RNAi with and without RU486), is sufficient to importantly decrease and restore the mRNA level (Fig. 8g-inset), and consequently increases (rescue) lifespan (Fig. 8g). Second, inversely, for the CG6296 (one of the most downregulated genes), to restore the level of its transcript, we constructed an UAS-cDNA and a transgenic fly-line. The targeted expression of the cDNA of CG6296 (Del,Mex-GS>pJFRC-MUH-cDNA-CG6396 with and without RU486), specifically in the gut and only in adulthood, restores the level of the mRNA (Fig. 8h-inset), and also leads to an important increase of lifespan. These results highlight a genetic and molecular link between the effect of the snoRNA on ribosomal RNA, some downstream deregulated genes and lifespan.

***Jouvence* modulates the resistance to various stresses**. Several stress-response genes have been reported to influence lifespan[5,14]. Moreover, several of the deregulated genes of the deletion revealed by the RNA-Seq analysis are also known to be involved in various stress resistances (e.g.: Glutathione and Cytochrome P450 family genes)[59]. Then, we wonder whether *jouvence* could affect resistance to stress. To elucidate *jou*'s role in stress response, we tested the resistance of transgenic lines (Del+rescue-1 and rescue-1) to desiccation and starvation. During the desiccation test, young flies were placed at 37 °C without water, and their survival time was quantified (Supplementary Fig. 8a). The median survival (50% of dead flies) of control and deletion flies was ~5.5 and ~4.5 h, respectively, showing that the *jou*-deleted flies were less robust. However, rescue-1 flies were more resistant, whereas Del+rescue-1 had similar lifespan as control, indicating that the genomic transgene rescued the resistance to stress. Since transgenic flies lived longer than controls, we also performed the desiccation test in aged (40-days-old) flies (Supplementary Fig. 8b). Similar results were obtained for old flies.

Next, the starvation assay revealed that the median lifespan of control and deletion was ~42 h and ~50 h (30% increase), respectively (Supplementary Fig. 8c). Rescue-1 flies were similar

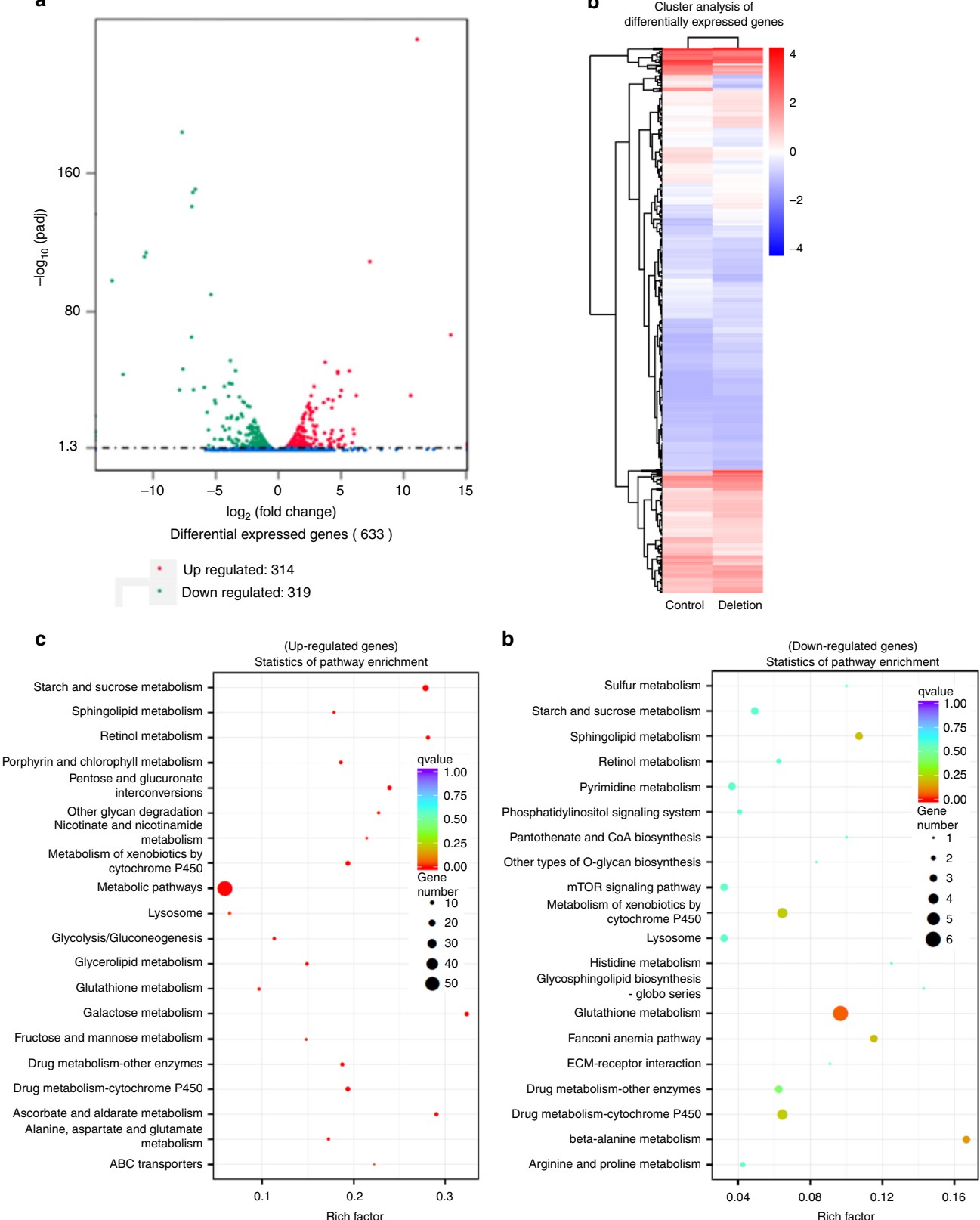

**Fig. 7 RNA-Seq analysis reveals that hundreds of genes are deregulated in deletion. a** Transcriptomic analysis (RNA-Seq) performed on total-RNA from the gut reveals that 314 genes are upregulated, while 319 are downregulated, in deletion compared to Control-CS (see Supplementary Data 1 and 2 for the complete list of genes). **b** Clusters analysis (Heat map) of differentially expressed genes. **c, d** Statistics of pathway enrichment of the deregulated genes according to the KEGG analysis[58] (see Supplementary Data 3 and 4 for the full list of the KEGG analysis). In brief, for the upregulated genes (**c**), the metabolic pathways are the main deregulated pathways, while for the downregulated genes (**d**), the glutathione metabolism is the main deregulated pathway (see the Supplementary Data 5 and 6 for the Gene Ontology Analysis).

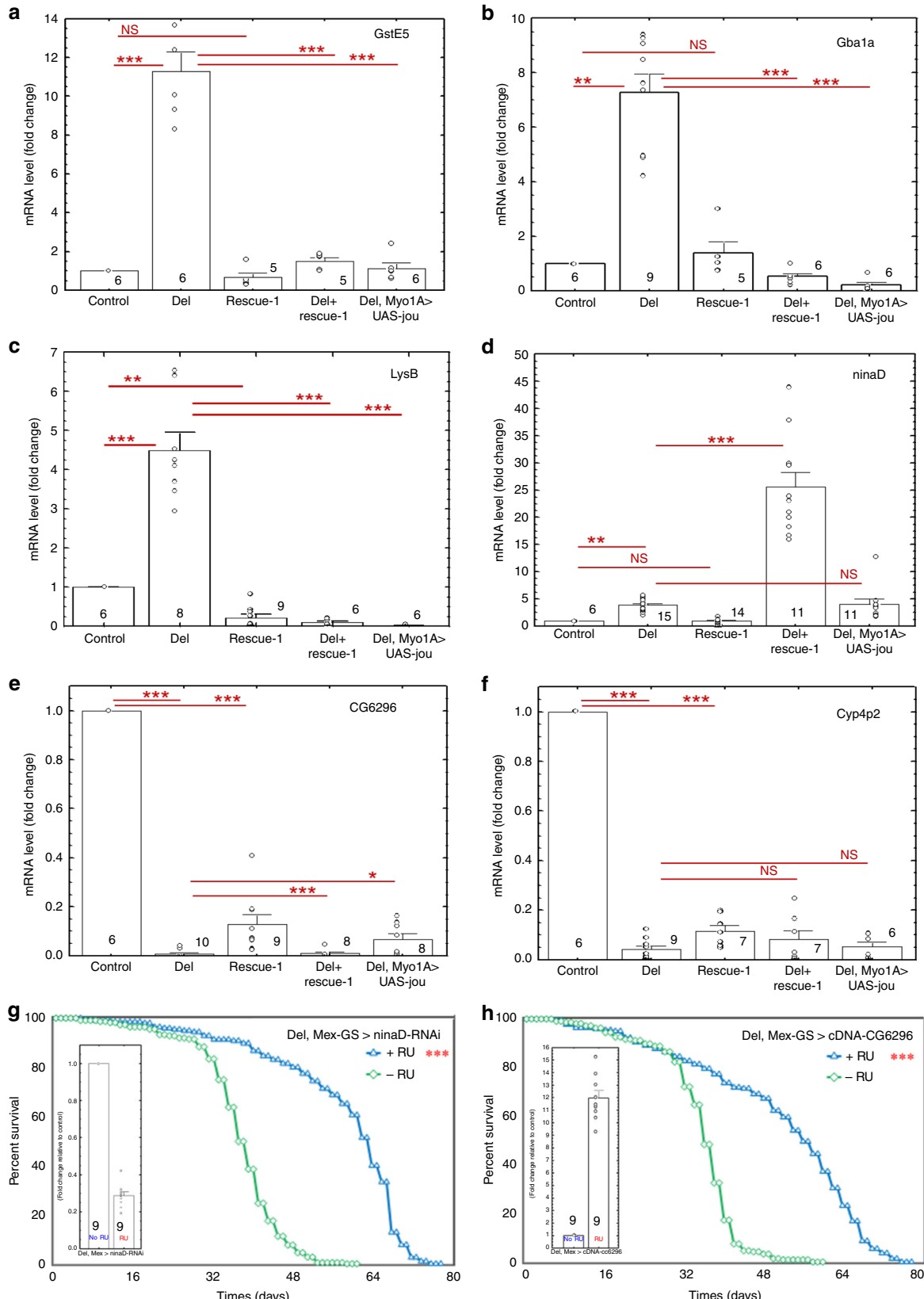

to control, while Del+rescue-1 had a higher lifespan than control, similar to deletion flies. In old flies, both control and deletion were less robust compared to transgenic overexpression (rescue-1) and Del+rescue-1 line (Supplementary Fig. 8d). These results

showed that *jou* deletion are less robust in desiccation but more robust in starvation in young flies, suggesting that *jou*'s protective role depends on the stress test. However, when *jou* expression was restored in enterocytes under the control of Myo1A-Gal4,

**Fig. 8 RT-qPCR confirms that genes are deregulated in deletion.** RT-qPCR (SybGreen) results of the quantification of the GstE5, Gba1a, LysB, ninaD, CG6296, and Cyp4p2 genes in Control (CS), Del (deletion F4), rescue-1, Del+rescue-1, and Del,Myo1A>UAS-jou[8M] in deletion genetic background. In deletion, the mRNA level is increased for GstE5, Gba1a, LysB, and ninaD genes, while in contrary it is decreased for CG6296 and Cyp4p2. For the genes GstE5 (**a**), Gba1a (**b**), and LysB (**c**), the mRNA level is restored in genomic-rescued transgenic flies Del+rescue-1 (even it is more than restored in LysB), while inversely for ninaD (**d**), the mRNA level is more increased than in deletion. The targeted expression of *jou* specifically in enterocytes (Del, Myo1A>UAS-jou[8M]) in deletion is sufficient to restore the level of mRNA for GstE5, Gba1a and LysB, but again here, not for ninaD. For the downregulated genes: CG6296 (**e**) and Cyp4p2 (**f**), none of the transgenic construction (Del+rescue-1 and Del,Myo1A>UAS-jou[8M]) rescues the level of mRNA (although some partial rescued is detectable with Del,Myo1A>UAS-jou[8M] for CG6296). **g, h** Longevity test results (survival curve—decreasing cumulative) of the targeted expression of UAS-ninaD-RNAi (**g**) and UAS-cDNA-CG6296) (**h**) specifically in the enterocytes in deletion genetic background compared to their controls. Restoring the level of ninaD (Del,Mex-GS>UAS-ninaD-RNAi) (**g**) only in adulthood (starting feeding flies with RU486 just after the flies have hatched) nicely increases lifespan compared to the non-RU486 fed control sibling flies. Similarly, restoring the mRNA level of CG6296 gene (Del,Mex-GS>UAS-cDNA-CG6296) (**h**) and so also only in adulthood, importantly increases lifespan compared to their control sibling flies. (*** = p < 0,001) (for number of flies, age in days at % mortality, and detailed statistics, see Supplementary Information, Table 1). RT-qPCR performed from the dissected guts of the RU486-induced only in adulthood (Del,Mex-GS>ninaD-RNAi or cDNA-CG6296), compared to the non-induced controls flies, confirm that the mRNA levels decreased (restored) for ninaD (inset **g**), and increased (restored) for CG6296 (inset **h**). For each panel, the level of the analysed gene has been normalized to the level of rp49. 30 guts were dissected for the total RNA extraction. The number at the bottom of the box (inside or just above) represents the number of samples (*n*). Statistics: RU486-induced compared to control non-induced (***p < 0,001). Errors bars represent the mean ± S.E.M. (*p*-value were calculated using the one-way ANOVA followed by a TUKEY test). For all panels, Source data are provided as a Source Data file.

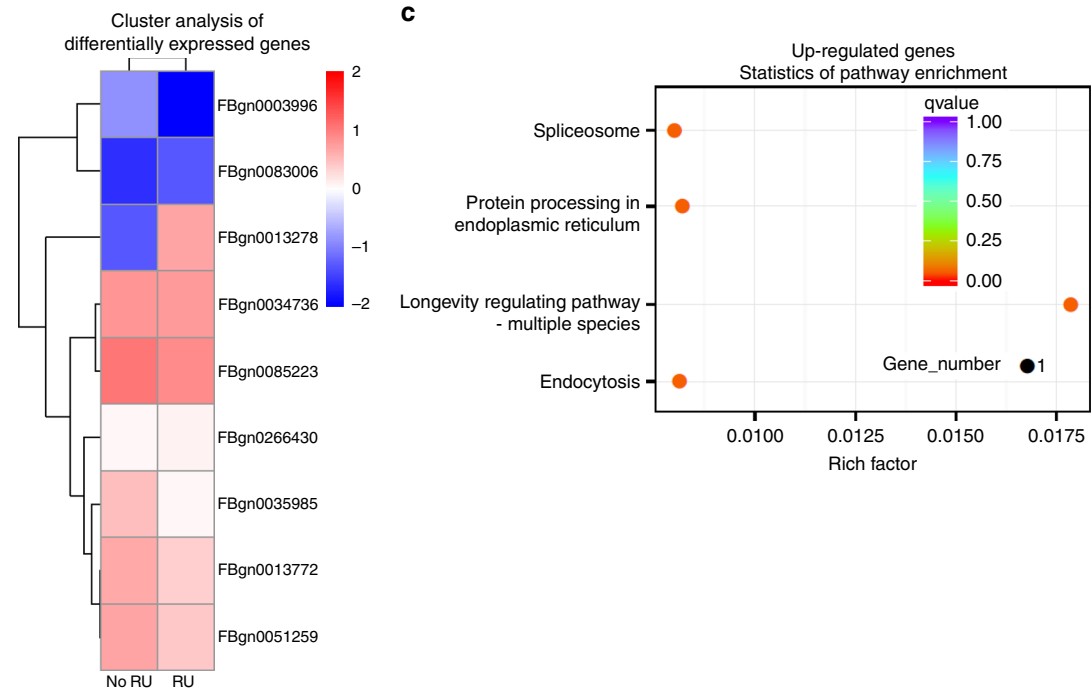

**a**

RNA-Seq on *jou* overexpressing flies (Mex-GS>jou[8M])

| Gene_id | readcount_MexG8RU | readcount_MexGSCo | Fold change | padj | Gene name | Gene description |
|---|---|---|---|---|---|---|
| FBgn0013278 | 2537,47 | 27,44 | 92,48 | 0,0000 | Hsp70Bb | Heat -shock-protein-70Bb |
| FBgn0266430 | 320,19 | 84,84 | 3,77 | 0,0000 | CG45060 | - |
| FBgn0013772 | 1460,79 | 718,11 | 2,03 | 0,0012 | Cyp6a8 | Cytochrome p450-6a8 |
| FBgn0051259 | 1882,84 | 940,11 | 2,00 | 0,0012 | CG31259 | - |
| FBgn0003996 | 222,34 | 82,33 | 2,70 | 0,0015 | w | White |
| FBgn0083006 | 27,65 | 0,00 | 0,00 | 0,0054 | snoRNA:Psi28S-1153 | - |
| FBgn0034736 | 2294,74 | 1006,53 | 2,28 | 0,0114 | gas | Gasoline |
| FBgn0085223 | 692,45 | 373,38 | 1,85 | 0,0483 | CG34194 | - |
| FBgn0035985 | 631,56 | 336,85 | 1,87 | 0,0485 | Cpr67B | Cuticular protein 67B |

**b** Cluster analysis of differentially expressed genes

**c** Up-regulated genes — Statistics of pathway enrichment

**Fig. 9 Only few genes are deregulated in flies overexpressing *jou*. a** Transcriptomic analysis (RNA-Seq) performed on total-RNA from the gut of flies overexpressing *jouvence* (Mex-GS>jou[8M]), RU486 fed flies (induced) compared to non-induced flies, reveals that only 9 genes are upregulated, while no gene are downregulated. **b** Clusters analysis (heat map) of differentially expressed genes. **c** Statistics of pathway enrichment of the deregulated genes according to the KEGG analysis[58] (see Supplementary Data 7–9 for the list of deregulated genes, list of KEGG genes, and Gene Ontology Analysis, respectively).

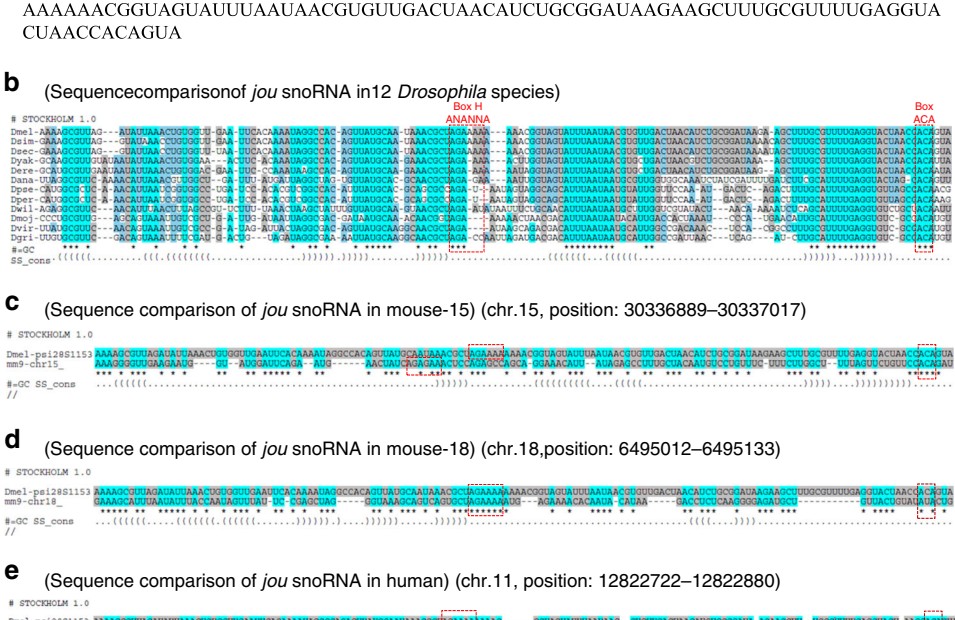

**Fig. 10 *jou* is conserved throughout evolution. a** RNA sequence of the *Drosophila jou* snoRNA. **b** Homology of *jou* sequences in 12 *Drosophila* species available on FlyBase. Dmel = *Drosophila melanogaster*, sim = *simulans*, sec = *sechellia*, yak = *yakuba*, ere = *erecta*, ana = *ananassae*, pse = *pseudoobscura*, per = *persimilis*, wil = *willistoni*, moj = *mojavensis*, vir = *virilis, gri = grimshawi*. **c, d** Ortholog sequences of *jou* in the mouse genome (*Mus musculus*). Two *jou* copies were detected, located on chromosomes 15 and 18. **e** Ortholog sequence of *jou* in the human genome. Only one copy was identified, located on chromosome 11. The canonical and conserved consensus sequence, as the H-Box (ANANNA) as well as the ACA-Box are highlighted (red boxes).

resistance to both desiccation (Supplementary Fig. 8e, f) and starvation (Supplementary Fig. 8g, h) was increased, supporting *jou* function in stress resistance. Together, these results indicate that *jou* is protective against the deleterious effects of stress.

**The snoRNA-*jouvence* is conserved through evolution**. The snoRNAs are known to be well conserved, both structurally and functionally, throughout evolution from archaebacteria to humans[22]. Bioinformatic analyses using BLAST and tertiary structure motif searches (using the Infernal software) revealed *jou* homologues in 12 *Drosophila* species, two orthologues in mice and one in human (Fig. 10), suggesting positive evolutionary pressure for this particular snoRNA. In mouse, putative *jouvence* snoRNAs are located on chromosome 15, at position: 30336889–30337017, and on chromosome 18 at position: 6495012–6495133, respectively. The unique putative snoRNA *jouvence* in human is located on chromosome 11, at position: 12822722–12822880. Moreover, in *Drosophila* species, the two other putative snoRNAs (sno-2 and sno-3), which are highly homologous (more than 90%) (Supplementary Fig. 9) (but completely different than *jouvence*) were identified just upstream to the snoRNA *jouvence* in 11 over the 12 other *Drosophila* species, except in *Drosophila grimshawi* (the most divergent from *D. melanogaster*), in which only the sno-2 is present. However, these two putative snoRNAs have not been identified upstream to the *jou* homologue in mammals, suggesting that these two snoRNAs are likely not directly functionally linked to *jouvence*.

To explore further a putative role of the snoRNA *jouvence* in mammals, we assessed expression by RT-PCR in various human and mouse tissues (Supplementary Fig. 10). In human, the unique putative *jou* snoRNA homologue was detected in the intestine, the brain, but weakly in ovaries and kidney. In mouse, snoRNA-

15 was detected in the brain, but not in ovaries, kidney and intestine, while the mouse snoRNA-18 was detected in the intestine, brain, ovaries, but not in the kidney (Supplementary Fig. 10a). Again here, to confirm and to provide a more direct measurement of the RNA size, we performed Northern Blots from mouse total RNA extracted from the gut, with the mouse-18 snoRNA, the one expressed in the gut. Supplementary Fig. 10b reveals a very faint band at 122 nucleotides corresponding to the expected size of *jouvence* (mouse-18). Finally, as snoRNAs are known to be generally located in large introns, we wonder if the genes in which the snoRNA is located or in the vicinity are expressed. In human, the snoRNA *jouvence* is located in a long intron of the TEAD1 (TEA domain family member 1) gene (Supplementary Fig. 11a). RT-qPCR reveals that the TEAD1 gene is expressed (though weakly) in the gut (Supplementary Fig. 12a). In mouse, the snoRNA-15 is located in the long intron of the Ctnnd2 (catenin delta-2: a cadherin associated protein) gene (also known as: Catnd2 or Nprap or neurojugin) (Supplementary Fig. 11b), which is weakly expressed (Supplementary Fig. 12b), while the snoRNA-18, which is located in the gene EPH1 (enhancer of polycomb homologue 1 isoform 1) (Supplementary Fig. 11c), is also expressed (though only faintly detected) (Supplementary Fig. 12 b).

**Discussion**

Here, we report the identification and the molecular characterization of a snoRNA (snoRNA:Ψ28S-1153) that we have named *jouvence* (*jou*), whose deletion decreases lifespan, while inversely, its overexpression increases it. Several independent genetic experiments support this conclusion. First, three independent genomic-rescued transgenic lines (Del+rescue-1; Del+rescue-2; and Del+rescue-3) rescue the longevity in deletion line. Second, since ISH revealed that *jou* was expressed in the enterocytes of the

epithelium of the gut, targeted specific expression of solely *jou* snoRNA in these specific cells, using two independent gut specific P[Gal4] driver lines (Myo1A-Gal4 and Mex-Gal4) and both combined to two independent UAS-*jouvence* transgenic constructs (UAS-jou[8M] and pJFRC-MUH-jou[1M]) were sufficient to rescue the longevity in the deletion, and even more precisely, only in adulthood (through the use of the Gene-Switch system) (Mex-GS). Moreover, Northern Blots analysis reveal that *jou* is detected in the gut at the expected size (148 nucleotides), validating its expression and processing. Finally, to extend our general characterization of the snoRNA-*jouvence*, RT-qPCR have been performed at different developmental stages revealing that *jouvence* is faintly expressed in all stages, except with a slight stronger expression in the white pupae (Supplementary Fig. 13), a result in agreement with the modENCODE (FlyBase)[35].

Transcriptomic analysis performed on deletion line has revealed that several (few hundred) genes are deregulated. Several of those genes have been validated by RT-qPCR, while the genetic rescue of *jouvence* specifically in the enterocytes (Myo1A-Gal4) was sufficient to rescue the mRNA level of several of them, demonstrating that *jou* is the snoRNA responsible for these deregulation. Moreover, among these genes, several are involved in stress resistance. Thus, while deletion line was more sensitive to stress, the rescue of *jou* in deletion increases the resistance to these stress, demonstrating that the presence of *jou* in the gut is sufficient to both increase lifespan and protect against various stress.

However, while our study was ongoing, a general transcriptomic analysis[35] has revealed two other putative snoRNAs, localized just upstream to the snoRNA:Ψ28S-1153 (*jouvence*). Since the DNA-fragment of 1727 bp used to build the genomic-rescue (rescue-1, 2, 3) also contains these two putative snoRNAs, thus a putative effect of these two snoRNAs could not be excluded. To verify and rule-out that the longevity effect was not due to these two putative snoRNAs, we have targeted the expression of the each of them (sno-2 and sno-3, respectively) in the enterocytes. The expression of the each of these snoRNA individually (in the deletion background) did not rescued or lead to an increase of longevity compared to their co-isogenic control lines, confirming that the longevity effect is due to the snoRNA-*jouvence*.

Our findings indicate that *jou* acts in the fly gut epithelium to promote longevity. In *Drosophila*, a relationship between lifespan and the gut have been previously reported[50,51,60]. In old Wild-Type flies, ISCs over-proliferation causes mis-differentiation of ISC daughter cells, resulting in intestinal hyperplasia: a phenomenon called "hyperproliferative diseases". Interestingly, preserving hyper-proliferation in the gut epithelium extends lifespan[51]. We have characterized both in control WT and deletion line, and so both in young and old flies, the number of different cell types in the gut. As reported, the number of ECs are increased in old WT flies, confirming a hyperplasia. Whereas, this increase is higher in *jou*-deleted flies, indicating more hyperplasia. However, remarkably, the targeted expression of *jouvence* in the gut, by two independent P[Gal4] driver lines (Del,Myo-Gal4>-jou[1M] and Del,Mex-Gal4<jou[1M]), as well as the overexpression (Myo1A-Gal4>jou[1M] and Mex-Gal4<jou[1M]) of *jou* prevents this hyperplasia. To point-out more precisely the origin of this hyperplasia, three different and independent quantitative analysis based either on a genetic marker (Dl-Gal4>UAS-GPF-nls), or immuno-histological staining (anti-PH3, EdU) have been performed and the labelled cells counted in the overall gut (Fig. 5). The number of ISCs is increased in deletion line (Del) compared to the Control-WT, as well as the number of dividing cells (anti-PH3 and EdU), three results that are consistent with the hyperplasia observed in deletion line. Considered altogether, these

results indicate a higher disruption of the proliferative homeostasis in deletion line, compared to their respective WT controls. More strikingly, the overexpression of *jou* either in deletion line or in the WT genetic background prevents the hyperplasia in old flies. Interestingly, this observation is in accordance with the decrease of the Cyclin E, a marker of the cell cycle, as well as with the decreases of the endoreplication in deletion line. Finally, we also check for the intestinal integrity of the flies, the so called "smurf" flies[11,16,60], at different ages (7–20 and 40-day-old flies), but we did not notice any significant difference between the Control-WT and the deletion flies (since in all of them, we observe almost no smurf flies, for about 100 flies for each group).

Recently, in *C. elegans*, a link between the nucleolus size and lifespan has been established[61], a mechanism that has been related to the nucleolar protein fibrillarin. The fibrillarin protein is associated to the RiboNucleoProtein (RNP) of the C/D box snoRNA, whereas the snoRNA *jouvence* belongs to the H/ACA box, which does not involve the fibrillarin. Nevertheless, in some ways, our study extends these findings[61] by specifically identifying a specific cell population (enterocytes) and a specific snoRNA (*jouvence*), reinforcing this previous finding that the nucleolus is involved in the determination of longevity.

Many longevity genes have been shown to modify the resistance to stress tests[5,14]. Interestingly, loss of *jou* function decreases resistance to desiccation, while inversely, the rescue or overexpression of *jou* increased resistance to this stress test. These protective effects are likely due to the increase of the Hsp78Bb gene expression level revealed by the RNA-Seq performed on overexpressing *jouvence* flies, a hypothesis that remain to be tested. However, in response to starvation, the *jou*-dependent resistance was found to be altered depending on the age of the fly (in deletion: it is increased in young flies, but decreased in old flies compared to Control). Such a difference and even opposite effect in resistance to various stress tests has previously been reported[28], suggesting that the effect on the stress resistance depends rather of the stress test itself rather than being a universal rule of the longevity genes. Indeed, each stress test relies on specific physiological and/or metabolic parameters that might be differently solicited for each stress. Dessiccation is mainly an indicator of the water physiology/homeostasis while the starvation is principally an indicator of the energy metabolism (lipids and/or glucose). In addition, since starvation effect of jou deletion changes during the adult lifespan (increased in young but reduced in old flies compared to controls), this could suggest that energy metabolic components are modified or altered during the course of the adult life.

To initiate deciphering the molecular mechanism of the snoRNA-*jouvence*, the RNA-Seq and RT-qPCR analyses revealed that several genes are either up- or down-regulated in the deletion line. Interestingly, the first striking outcome of the overview of the 633 differentially expressed genes (Supplementary Data 1–7) reveals that the main components of the most well-characterized pathways involved in longevity, the insulin-signalling pathways (*InR*, *chico*, *FOXO*, etc.), are not strikingly affected, except the *dILP3* (Supplementary Data 2, line 157) which is downregulated in the deletion line, while the *dILP4* (Supplementary Data 1, line 86) is upregulated. These results suggest that *jou* could act through mechanisisms or pathways related to longevity, or acts as a more general mechanisism to regulate the expression level of several genes. In such a way, the KEGG analysis has revealed that several physiological pathways are affected, the main one being the metabolic pathway with 55 genes (over 924) deregulated. The second striking observation revealed by the RNA-Seq concerns the reported link between the Myc/ribosomes biogenesis and lifespan, either in *C. elegans*[62], *Drosophila*[63], and mouse[64]. Indeed, it has been reported that several snoRNAs are a class of

Myc targets[65]. However, though this study conducted on S2 cells has identified about 500 Myc target genes, whose function mainly resides in the control of ribosome biogenesis, the snoRNA *jouvence* itself has not been identified in this group of genes regulated by Myc. This result is corroborated by the fact that we did not found any Myc consensus E-box sequence (CACGTG)[65] in the vicinity of the *jouence* gene (the nearest one being located at 3112 bp in 5'), as well as the Myc gene and the ribosomes biogenesis pathways are not affected in deletion line (see DEG, Supplementary Data 1, 2 and KEGG analysis, Supplementary Data 3). Therefore, this *jou*-independent Myc regulation rather leads to hypothesis that *jouvence* may act, through the pseudouridylation, on ribosomes stability and/or optimize the ribosomes function (which argues for the concept of specialized ribosomes)[66] instead of acting directly on ribosome synthesis.

Moreover, some of these genes have been validated by RT-qPCR. In addition, for some of them (*GstE5*, *Gba1a*, and *LysB*), the level of RNA was rescued by the re-expression of *jouvence* in the enterocytes, demonstrating that *jou* is clearly the snoRNA responsible of this deregulation. However, for some other genes (*ninaD*, *CG6296* and *Cyp4p2*) the deregulation seems to be more complex, either requiring a very precise level of snoRNA-*jou*, or potentially involving the two other putative snoRNAs. To identify and confirm a link between the snoRNA-*jouvence*, the deregulated genes, and the longevity phenotype, we restore in the deletion genetic background, for two of them (*ninaD* and *CG6296*), the level of their transcripts. Remarkably, decreasing the level of *ninaD* (neither inactivation nor afterpotential D) by the expression of a RNAi specifically in the gut, or inversely increasing the level of *CG6296* (a downregulated gene) by the expression of its cDNA, increase lifespan, demonstrating that these two genes, even taken individually, are crucial for the determination of lifespan (Fig. 8). However, these two rescues of longevity is not a systematic rule for all of deregulated genes, since we have also analysed the Gba1a gene (Del,Myo-Gal4>UAS-Gba1a-RNAi) (Supplementary Fig. 15), but this last did not restore the longevity of the flies. Interestingly, based on their sequence homology, these two genes are predicted to be involved in lipids and/or cholesterol metabolism. Indeed, *ninaD* encodes a membrane protein homologous to the mammalian class B type I Scavenger receptors, SR-BI. In Mammals, SR-BI play critical roles in cholesterol and high-density lipoproteins (HDL) metabolism and in maintaining plasma cholesterol levels. CG6296 (a non yet well characterized gene) presents protein domains/motifs similar to triacylglycerol lipase family, with predicted lipase activity and phosphatidylcholine 1-acylhydrolase activity, suggesting a role in lipids metabolism (FlyBase). Taken together, these results indicate that the snoRNA *jouvence* is involved in the fine tuning of the level of many mRNAs. We have demonstrated that *jouvence* pseudouridylates the 28S-rRNA and 18S-rRNA as predicted by its primary sequence[34], a function restored by the various transgenic lines. Therefore, the snoRNA-*jouvence* might likely act either in the maturation and/or stabilisation of the rRNA, although an involvement in the regulation of the transcription or even potentially in the chromatin structure, as already reported for some other snoRNAs[25] could not at this stage of our study, be excluded. Interestingly, as reported for the methylation of ribosomal RNA by NSUN5, in which the modification of a single rRNA nucleotide yields to functionally different ribosomes[66], here, the pseudouridylation of a specific base on rRNA ($\Psi$ at position 1153 on 28S-rRNA) potentially generates a specialized ribosome. Further experiments, as for example, the determination of the 3D structural conformation of the ribosomes (pseudouridylated or not) will be required to answer this precise question. Finally, although it is commonly considered that the snoRNAs might mainly be an ubiquitous and housekeeping machinery in all cells, obviously, as for the methylation of

ribosomal RNA by NSUN5[66], our results rather argues for a specific snoRNA machinery in certain tissues, as for instance here, in the epithelium of the gut.

Based on its primary sequence (Fig. 10a), *jou* can be classified in the box H/ACA snoRNA[22]. Interestingly in humans, some pathologies have been associated with various snoRNAs. For instance, a mutation in the H/ACA box of the snoRNA of the telomerase (a RNP reverse transcriptase) yields a pleiotropic genetic disease, the congenital dyskeratose, in which patients have shorter telomeres[26,27]. It has also been reported that the snoRNA HBII-52, a human C/D box type of snoRNA, regulates the alternative splicing of the serotonine receptor 2C[67]. Therefore, we speculate that the mammalian *jou* orthologue could play important functions in mouse and human respectively, but it remains to be investigated. Nevertheless, the conservation of sequence and the expression of *jou* raise the exciting hypothesis that similar functions may also be conserved in vertebrates including humans.

Thus, undoubtedly, *jou* represents a good candidate to understand the relationship between the gut, the aging, and the longevity. Interestingly, as already reported, the intestinal cells are very similar in *Drosophila* and mammals, both at cellular and molecular levels[68], while snoRNAs are well conserved throughout evolution[22]. Although *jouvence* was not yet annotated in the mammalian genome including human, we have shown by two independent and complementary approaches (RT-PCR and Northern Blot) that *jou* is present in mammals and is expressed in the gut (among other tissues), strongly suggesting that it might be functional. Therefore, we could hypothesize that manipulating the mammalian gut epithelium may also protect against deleterious effects of aging and even increase longevity (including in humans). Thus, *jou* snoRNA could represent a promising therapeutic candidate to improve healthy aging.

## Methods

**Flies.** *Drosophila melanogaster* lines were maintained at 24 °C, on our standard dried brewer yeast and cornmeal medium (without sugar or molasses), in a 12 h light/12 h dark cycle. Canton Special (CS) flies were used as Wild-Type control. The deletion (F4) (short named: Del) and the genomic rescue transgenic flies (rescue-1, -2, -3, Del+rescue-1; Del+rescue-2; and Del+rescue-3) were outcrossed a minimum of 6 times with CS (Cantonization) to thoroughly homogenize the genetic background. To follow the deletion (F4) during the Cantonisation (and similar for the Berlinisation), PCR has been performed at each generation to re-identify the fly (chromosome) bearing the mutation (genotyping). The Gal4 lines used (Myo1A-Gal4, Mex-Gal4, esg-Gal4, Dl-Gal4, and the generated Mex-GS), as well as the generated transgenic lines (UAS-jou[8M], p-JFRC-MUH-jou[1M], UAS-sno-2, UAS-sno-3) were also backcrossed 6 times with CS (Cantonization). The two Gal4 lines (Myo1A-Gal4 and Mex-Gal4) were then crossed to each of the snoRNA transgenic construct lines to determine their longevity, while in parallel, they were crossed to deletion (F4) to generate the co-isogenic control flies (in such a way, all the tested lines were in heterozygous both for the Gal4 insertion as well as for the UAS-transgene). The lines Myo1A-Gal4 (located on chromosome 2) and esg-Gal4 (chromosome 2), were kindly provided by B.A. Edgar, Heidelberg, Germany. The lines Su(H)GBE-Gal4 (chromosome X), and Dl-Gal4 (chromosome 3) were a courtesy of X. Zeng, NCI/Frederick, USA. Mex-Gal4 (chromosome 2) was kindly provided by G.H. Thomas, Pennsylvania State Univ., PA, USA. The UAS-ninaD-RNAi line (31783R) has been provided from NIG (Japan). This line has also been backcrossed 6 times with CS (Cantonization), and thereafter introduced in the deletion background by standard genetic crosses.

**Determination of the deletion (F4) by PCR.** Two primers located at positions 66851F and 68140R (see Supplementary Information for the sequence of all primers) have been used to amplify the genomic DNA region, both in WT-CS and deletion, to reveal the deletion. The amplicon in CS is 1289 pb, while in deletion is 657 pb. In complement, the deletion has been sequenced. The same two primers have been used to follow the deletion during the Cantonisation and the Berlinisation.

**Construction and genesis of the DNA genomic rescued lines.** The *jou* genomic transgenic lines were generated by PCR amplification of a 1723 bp genomic DNA fragment, using the forward primer (snoRNAgenomic-F) and the reverse primer (snoRNAgenomic-R) (see Supplementary Information for the sequence primers),

both adding a *XbaI* cloning site at the 5'-ends. The amplified fragment was then inserted via the Xba1 restriction site into the pCaSper 4 (which contains no enhancer/promoter sequence). This way, snoRNA expression is dependent on its own genomic regulatory sequences included in the inserted DNA fragment. The transgenic flies were generated by standard transgenic technique (Bestgene, USA), and three lines have been obtained (rescue-1, rescue-2, and rescue-3). Rescue-1 and rescue-2 are inserted on the third chromosome, while the rescue-3 is inserted on the second chromosome.

**Construction and genesis of the Mex-Gene-Switch line**. To generate the Mex-Gene-Switch line, we use the forward primer: Mex1-F + a Sfi1 cloning site at the 5'-ends: 5'-ATAGGCCGGACGGGGCCAACGCGAATTCAGACTGAGC-3', and the reverse primer: Mex1-R + a Sfi1 cloning site: 5'-TGTGGCCCCAGTGGCC CGTTGCACATGGTGATGACT-3', to amplify by PCR using the genomic DNA, a fragment of 2264 bp. Then, this DNA fragment containing the regulatory/promoter sequence of the Mex gene, as formerly reported[47] has been inserted through the EcoR1 and DraII sites into the pattB-Sf-HS-GALGS (Sfi1 orientated) (kindly provided by H. Tricoire, Paris). The transgenic flies have been generated by BestGenes (USA), in which the plasmid-construct has been inserted in VK02 site located at position 28E7 on the second chromosome. The Mex-GS transgenic line has been re-introduce in the deletion background by standard genetic cross and recombination.

**Genesis of the UAS-snoRNA lines (*jou*, sno-2, sno-3)**. First, the snoRNA *jou* of 148 bp was cloned into the TOPO-TA cloning vector (Invitrogen) using the same primers used for RT-PCR (see Suppl. Information for snoRNA-Fb and snoRNA-Rb). This construct was also used to synthesize probes for ISH. First, the snoRNA fragment of 148 bp was subcloned into the p[UAST] vector. Second, similarly, the snoRNA fragment of 148 bp was subcloned into the pJFRC-MUH vector[49] (a gift from Gerald Rubin: Addgene plasmid #26213; http://n2t.net/addgene:26213; RRID: Addgene_26213) containing the attB insertion sites allowing a directed insertion. Similarly, for the cloning of the DNA fragment containing the sno-2 or the sno-3, appropriated forward and reverses primers (see Supplementary Information) were used to amplify a DNA fragment of 166 pb for the sno-2, and 157 pb for the sno-3, respectively. Each fragment was then cloned directly into the pJFRC-MUH vector. The transgenic flies were generated (BestGenes, USA). The p[UAS]-jou[8M] is inserted randomly on the third chromosome, the pJFRC-MUH-snoRNA-jou[1M] construct was inserted into attP2 site located at position 68A4 on the third chromosome, while the sno-2, and the sno-3 were inserted in VK27 site located at position 89E11 on the third chromosome.

**Construction and genesis of the pJFRC-MUH-cDNA-CG6296**. To generate a Gal4 inducible UAS-cDNA-CG6296, as the framework, we use the CG6296 cDNA cloned in the pOT2 vector (reference: LP07116 from BDGP resources). We then use the forward primer: CG6296F + a BglII cloning site at the 5'-ends: 5'-TTCACT TCAACGATGAGATTGAGATCT-3', and the reverse primer: CG6296R + a XbaI cloning site: 5'-TCTAGAGGACTTGCAATATTATTTGA-3', to amplify by PCR a fragment of 2167 bp. Then, this cDNA fragment has been inserted through the BglII and XbaI sites into the pJFRC-MUH vector. The transgenic flies have been generated by BestGenes (USA), in which the plasmid-construct has been inserted in VK27 site located at position 89E11 on the third chromosome. Thereafter, the pJFRC-MUH-cDNA-CG6296 transgenic line has been backcrossed 6 times with CS (Cantonization), and then introduced in the deletion background by standard genetic crosses.

**Detection of *jou* by RT-PCR in *Drosophila*, mouse and human**. For *Drosophila*, total RNA from whole female flies was extracted by standard procedures using Trizol (Invitrogen). This was treated with RQ1 RNase free DNase (Promega) to remove any DNA contamination, and verified by PCR. Following this, RT was performed with MMLV Reverse Transcriptase (Invitrogen) using a random primer hexamer (Promega), followed by a 20-min RNase H (Invitrogen) at 37 °C. The PCR was performed using forward and reverse primers (for primer sequences, see Supplementary Information), with Taq DNA Polymerase (Invitrogen), at 51.4 °C. Similar procedures were used to amplify the 300 bp control RP49 fragment using the forward and reverse primers respectively. Mouse and human total RNA were commercially purchased at Amsbio: Mouse Brain, whole Total RNA: MR-201, Mouse Ovary Total RNA: MR-406, Mouse Kidney Total RNA: MR-901, Human Brain Total RNA: HR-201, Human Ovary Total RNA: HR-406, Human Kidney Total RNA: HR-901. The human gut RNA was purchased at Biochain: Human adult Normal small Intestine Total RNA: T1234226. All of these total RNA were re-treated by the DNAse to completely remove the DNA, and re-verified by PCR before performing the RT-PCR. Thereafter, RT-PCR have been performed in similar conditions as for *Drosophila*. For the sequence primers, see Supplementary Information.

**Quantification of genes expression by RT-PCR**. For *Drosophila*, total RNA were extracted from 30 dissected gut as described above for the detection of *jou* expression, followed by RT-PCR. For the *Drosophila* Cyclin-E, see Supplementary Information for the primers used. rp49 was used as normalization. For the mouse

and human RT-PCR, we use the same total RNAs as for the *jouvence* orthologue detection (see above). For the TEAD1 (human), CNNT and EPC1 (mouse), see Supplementary Information for the primers used. Human and mouse 18S-rRNA were used as normalization, accordingly. All the genes expression were performed by RT-qPCR using SybrGreen probes.

**Quantitative PCR (qPCR) by Taqman**. For RNA extraction of whole flies or dissected gut, similar procedures described above for RT-PCR were used. The samples were diluted in a series of 1/5, 1/10, 1/100 to establish the standard curve, then 1/100 was used for the internal control (RP49) and 1/5 for *jou*. The TaqMan Universal Master Mix II, no UNG was used with a specific Taqman probe designed for the snoRNA *jouvence* and a commercial Taqman probe for RP49 (catalogue number: 4331182, ThermoFisher[TM]). The amplification was done an a BioRad apparatus (Biorad CFX96) with a program set at: activation 95 °C, 10 min, PCR 40 cycles: 95 °C, 15s, 60 °C, 1 min. Then, the ratio of the snoRNA (dilution 1/5) was reported to the RP49 (dilution 1/100 - since the RP49 is more expressed) and then calculated and normalised to CS = 1. Taqman RT-qPCR was performed in duplicate on each of 3 or 5 independent biological replicates, depending of the genotype. All results are presented as means ± S.E.M. of the biological replicates.

**In situ hybridization (ISH) and histological preparation**. For whole flies ISH, we used 5 day-old females. On day-1 (d1), whole flies were fixed with 4% paraformaldehyde (PFA) for 6 h at 4 °C, and then incubated overnight in 25% sucrose solution at 4 °C. On d2, individual flies were embedded in 20% carboxymethylcellulose, frozen in liquid nitrogen, and sectioned using a Cryostat (−20 °C) at 30 μm thickness. The sections were rehydrated for 15 min in PBS-1X + 0.1% Tween-20, and post-fixed with 4% PFA for 15 min. They were washed twice with PBS-1X + 0.1% Tween-20, and treated with Proteinase-K (10 μg/ml) for 5 min. Proteinase-K was inactivated using glycine (2 mg/ml) and the sections were washed and post-fixed again. For ISH, the fixed sections were washed with hybridization buffer (50% formamide, 5X SSC, 100 μg/ml heparin, 100 μg/ml of salmon sperm DNA, 0.1% Tween-20) followed by pre-hybridization with PBS (1/1) diluted by PBS (1/1) at 65 °C using the same hybridization buffer. Overnight hybridization with the snoRNA anti-sense probe (or sense as negative control) was performed at 45 °C. The sense and anti-sense probes were synthesized using the DIG RNA labelling kit (SP6 or T7) (Roche[TM]). On d3, the sections were washed four times at 45 °C: 3/0 - 3/1 - 1/1 - 1/3 with hybridization buffer/PBS, 15 min each wash. Sections were then washed four times with PBS 1X 0.1% tween, 5 min at RT, and PBS 1X 0.1% tween + 10% goat serum, 1 h, followed by the incubation with the primary antibody, anti-Digoxigenin-mouse-HRP conjugate at 1/100 (reference: NEF832001EA, Perkin Elmer) for 1h, RT. The immuno-reaction was amplified with the tyramide amplification kit (TSA cyanine 3 plus Evaluation kit—Perkin Elmer) for 8 min in the dark; the sections were washed as usual and counter-stained with DAPI (Roche) for 5 min. The preparation was mounted with Mowiol. For double labelling, following tyramide signal amplification, the sections were incubated with goat serum 2% in PBS 1X 0.1% Tween-20, for 2 h at RT, and incubated with anti-GFP at 1/500 (reference: 11-814-460-001, Roche), overnight. On d4, the slides were washed, and incubated with the secondary antibody, anti-mouse FITC at 1/500 (F-6257, Sigma), washed as usual, and mounted with Mowiol. Same protocol has been used for the ISH on dissected gut, except that we use a FITC-labelled tyramide instead of a Cy3-labelled tyramide.

**Northern blot**. Total RNAs from the different genotypes were extracted from the dissected gut with TRIzol reagent following the supplier's instructions (Invitrogen). Denatured RNA samples were run for 1 h on precasted (10 × 10 cm) TBE urea 10% acrylamide gel (Biorad). We use, for *Drosophila*, 10 μg of total RNA, and for mouse, 15 μg of total RNA extracted from the gut. The RNA were then transferred to Hybond N+ membranes (Amersham) and then pre-hybridized for 4 h in 50% formamide, 5X SSC, 5X Denhardt, 50 mM sodium phosphate pH 7.5, and 0.1% SDS (w/vol), and 100 μg ml$^{-1}$ of denatured DNA from salmon sperm, before an overnight hybridization under the same conditions. Pre-hybridization and hybridization were carried out at 42 °C. The day after, the membrane were washed as follows: a spontaneous wash with 4X SSC during 5 min at RT, followed by three successive washes of 4X SSC, 0.1% SDS, 1X with 1X SSC, 0.1% SDS, and finally 1X with 0.5X SSC, 0.1% SDS, each during 20 min at 60 °C. Membranes were then subjected to phosphoimager analysis using a Typhoon Trio Imaging System (GE Healthcare) for 2 h. For the P$^{32}$ labelled RNA probes, first, the DNA fragment for each probe (*Drosophila jouvence* and U6, mouse snoRNA-18 and U6) were cloned in the TOPO-TA Cloning plasmid vector (Invitrogen). Then, according to the sens of the cloning, the 3' end of each probe were cut with the appropriated restriction enzyme. Thereafter, the transcription of the probe in presence of α-P$^{32}$ UTP were synthethized using either SP6 or T7 RNA polymerase. Finally, the probes were purified on Sephadex spin-column. In a second step, Blots were re-probed with U6 as a loading control.

**Pseudouridylation**. Total RNAs from the different genotypes were extracted from the dissected gut with the SV Total RNA Isolation System Kit (Z3100, Promega), followed by a DNase-1 treatment (30 min at 37 °C) to completely remove any DNA trace. Then the total RNA were purified with the NucleoSpin RNA Clean-up kit

(740948.50, Macherey-Nagel), and the concentration determined. For pseudouridylation experiments, we follow the protocol of Huang et al. (2016)[69]. Dried total RNA (10µg for each reaction) were treated with (or without for control) N-cyclohexyl-N'-(2-morpholinoethyl)-carbodiimide metho-p-toluenesulphonate (CMC) for 30 min at 40 °C, and subjected to alkali hydrolysis in the presence of 50 mM $Na_2CO_3$ (pH 10.4) at 50 °C during 2 h with shaking every 10 min. The primers of each probes (18S and 28S-rRNA) were radiolabeled using the T4 Polynucleotides Kinases (T4-PNK) and γ-P$^{32}$-ATP. The 20 mers primers (see Supplementary Materials for the sequences) were designed about 50 nucleotides downstream to the predicted sites of pseudouridylation. The Reverse Transcription reactions were then performed on the 18S and 28S-rRNA templates respectively. In meantime, the DNA templates of the 18S and 28S-rRNA were generated by PCR using two primers located about 200 nucleotides upstream and downstream to the predicted pseudouridylation sites (a fragment of about 400 bp) (see Suppl. Materials for the sequences). For the sequencing of the DNA used as reference, we used the Sanger method and the Sequenase Kit (USB, Life Technologies), with slight modifications, notably because we used γ-P$^{32}$ 5'-labelled primers (the same as for the pseudouridylation reaction) instead of labelled nucleotides. Then, both reactions were migrated in parallel on an 8% acrylamide gel + 8 M urea, at 2000 volts during 2 h. After drying, the gels were then subjected to phosphoimager analysis using a Typhoon Trio Imaging System (GE Healthcare) for 2 h.

**Determination of the cells number of different cell types**. First, to determine the number of cells of the different cell types, the gut were dissected in PBS and then fixed in 4% PFA for one hour, followed by 3 washes with PBS and then stained with DAPI for 5 min, and finally mounted in Mowiol. To determine the number of ECs, we count the total number of nucleus stained with DAPI on microscope images taken on two distinct areas of a 33,000 µm$^2$ in the posterior midgut. The displayed data represent the means of these two distinct areas. Second, the number of each other specific cell types were counted on the whole gut. We used appropriated genetic markers: for the ISCs, we use Dl-Gal4 driving the UAS-GFP-nls (nls = nuclear localization sequence). For the dividing cells, we use two independent approaches (EdU and anti-PH3). For the EdU labelling, the guts were dissected in the PBS and incubated for 1 h in EdU solution (20 µM) at 25 °C (according to the protocol of Lilly and Spradling, 1996)[53], using the Click-iT Reaction Imaging Kit (C10086, Molecular Probes). The gut were then mounted in Mowiol and examined to the microscope. For the anti-phospo Histone-H3 (anti-PH3) immunostaining, the guts were dissected in the PBS and fixed for 1 h in 4% PFA. Then washed 3 times in PBST (PBS + 0,1%Triton), followed by 4 h incubation in PBST, and then with the primary antibody (polyclonal anti-phospho-Histone-3 done on rabbit) at 1/2000 (EMD-Milipore, #06-570) overnight at 4 °C. The day after, the gut were washed 3 times, followed by one hour incubation with the secondary antibody (goat anti-rabbit-Alexa488) at 1/400 (A11304, Invitrogen) at RT, then re-washed again 3 time with PBST and then mounted in Mowiol.

**Quantification of endoreplication**. To quantify the endoreplication of the enterocytes, we used the same fixation protocol as to count the number of enterocytes, except that we precociously and strictly quantify the DAPI-fluorescence according to Weiss et al., 1998[56].

**Longevity**. Following amplification, flies were harvested on one day. Female flies were maintained with males in fresh food vials for 4 days. On day 4, females were separated from males, and distributed in a cage holding between 200 and 250 females. The cage (50 mm diameter × 100 mm high) is made of plexiglass. The top is covered by a mesh, while the bottom (floor) is closed by a petri dish containing the food medium. The petri dish was replaced every 2 days, and the number of dead flies counted. Note: The lifespan determination of each group of flies and their respective controls (genotypes) have been performed simultaneously, strictly in parallel. Moreover, all longevity experiments have been repeated 2 or 3 times independently.

**Stress resistance tests**. For starvation test, female flies were maintained with males in fresh food vials for 4 days, before testing. On day 4, females were separated from males, and distributed in vials containing 20 females each. Each vial contained a filter, and cotton moisturized with 200 µl of water, to keep the filter moist during the test. The flies were kept at 24 °C in a humid chamber, to avoid dehydration. Dead flies were counted every 12 h.

For desiccation test, flies were reared at 25 °C, as usual. Then, female flies were maintained with males in fresh food vials for 4 days, at 25 °C, before testing. On day 4, females were separated from males, and distributed in empty glass vials containing 20 females each, without any filter paper. The vials were transferred and tested at 37 °C, for young flies. However, since 40-day-old flies die rapidly at 37 °C precluding a clear differentiation of the various genotypes, the stress test assay was performed at 30 °C, a milder stress condition. Dead flies were counted every hour.

**Transcriptomic analysis (RNA-sequencing)**. For RNA extraction, each single hand-dissected gut was immediately soaked in liquid nitrogen and keep at −80 °C. 30 guts were dissected for each genotype. RNA extraction was performed with SV Total RNA Isolation kit (Wizard Purification, Promega) following standard

procedures provided by the manufacturer. Thereafter, the total RNA was treated with RQ1 RNase free DNase (Promega) to remove any DNA contamination, and verified by PCR. Then, total RNA were checked for integrity on a bioanalyzer (Agilent). PolyA RNA-Seq libraries were constructed using the Truseq mRNA stranded kit (Illumina) according to the manufacturer recommendations. Libraries were sequenced (Single Read 75 pb) on an Illumina NextSeq500 instrument, using a NextSeq 500 High Output 75 cycles kit. Demultiplexing has been done (bcl2fastq2 V2.15.0) and adapters removed (Cutadapt1.9.1). Reads have been mapped on the *D. melanogaster* genome (dmel-all-chromosome-r6.13.fasta) with TopHat2. Mapped reads were assigned to features with featureCounts 1.5.0-p2 and differential analysis has been done with DESeq2. Three independent biological replicates have been done for each genotype. For the Wild-Type CS and deletion (F4), the RNA-Seq has been performed in the Platform of the I2BC, CNRS, Gif-sur-Yvette, followed by a Bioinformatic analysis performed by Novogene Co (Beijing, China). For the overexpression of *jouvence* (Mex-GS>UAS-jou$^{8M}$) flies induced by feeding with RU486 (during 7 days) have been compared to non-induced flies (without RU486 feeding). For this last, the complete RNA-Seq analysis (Sequencing and Bioinformatic analysis) has been performed by Novogene.

**Quantitative and statistical analysis**. For the longevity and stress resistance survival curves, log-ranks test was performed using the freely available OASIS software (https://sbi.postech.ac.kr/oasis2/)[70]. For the qPCR, and the quantification of number of different intestinal cells, data were analysed statistically using analysis of variance (one-way ANOVA) followed by a TUKEY test, with Statistica$^{TM}$ software.

We declare to have complied with all the relevant ethical regulations for animal testing and research.

**Reporting summary**. Further information on research design is available in the Nature Research Reporting Summary linked to this article.

## Data availability

The authors declare that all the data and the methods used in this study are available within this article, its Supplementary Information files, its Supplementary Data, its Source Data File, or are available from the corresponding authors upon reasonable request. The RNA-Sequencing Data have been deposited at NCBI with the BioSample accession number: PRJNA594829.

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

## Acknowledgements

We thank E. Heard for her help with the search of the mammalian orthologue sequences. F. Rouyer (NeuroPSI, Gif-sur-Yvette), A. Molla-Herman, J.R. Huynh, and more especially A. Bardin (Institut Curie, Paris) for their helpful comments and critical reading of the paper. B.A. Edgar (Heidelberg, Germany), X. Zeng (NCI/Frederick, USA), and G.H. Thomas (Penn. State Univ., USA) for the gut-Gal4 lines, H. Tricoire for the pattB-Sf-HS-GALGS (Gene-Switch vector), and M.N. Soler from Imagif CNRS for her assistance with confocal microscopy. We also thank M. Betermier and her Team (CNRS, I2BC, Gif-sur-Yvette) for the loan of sequencing/pseudouridylation materials. This work has benefited from the facilities and expertise of the high throughput sequencing core facility of I2BC (Centre de Recherche de Gif - http://www.i2bc-saclay.fr/). The English language of the preliminary version of this paper has been edited by ASK Scientific Ltd, Cambridge, UK, and by A. Bardin for the last version. This work was supported by the CNRS France to J.R. Martin.

## Author contributions

J.R.M. conceived and designed the experiments. J.R.M., S.S., L.M. and K.A. performed Drosophila experiments and analysed data. C.C. performed bioinformatics search of jou orthologues. J.R.M. wrote the paper with input from all authors.

## Competing interests

The authors declare no competing interests.

**Additional information**

