## [Peer Review File · Nature Communications]

Reviewers' comments:

Reviewer #1 (Remarks to the Author):

This manuscript describes the interesting observation that the levels of a small nucleolar RNA (*jou*) modulate lifespan from intestinal enterocytes in adult *Drosophila*. In a series of carefully controlled experiments, the authors show that *jou* is specifically expressed in enterocytes and ovarian cells. Flies lacking *jou* are viable but shorter-lived. One might argue that a shorter lifespan is only to be expected from losing a “housekeeping” function but, strikingly, *jou* overexpression exclusively in adult enterocytes is sufficient to increase lifespan substantially. The experiments provided are unusually well controlled; flies have been backcrossed for several generations to two different reference stocks, several overexpression lines have been used, a possible contribution of two neighbouring genes has been ruled out, two different enterocyte drivers have been used and *jou* expression levels have been carefully quantified in both loss- and a gain-of-function experiments. Mechanistic exploration of how *jou* modulates lifespan includes transcriptional profiling of guts lacking *jou* relative to controls, and description of the effects of *jou* absence on desiccation/starvation resistance. Although such analyses are relatively superficial, the initial observation is novel and well documented. The authors also perform computational analyses that have revealed mammalian orthologues of *jou*. Their initial expression analyses indicate that at least one of them is expressed in the mouse intestine, raising the possibility that their *Drosophila* findings are more broadly significant.

Specific comments

1. The authors need to mention Tiku et al 2016 (PMID: 28853436). Although their manuscript extends the findings described by Tiku et al in various ways (they show a role in flies rather than worms, they identify a specific cell population and a specific snoRNA involved), it is important to put these findings in the context of previous links between the nucleolus and lifespan. The authors may also want to discuss their findings in the context of known links between *Myc*/ribosome biogenesis and lifespan, and known correlations between reduced ribosomal protein content and increased lifespan in worms.
2. The cell type specificity of *jou* expression is striking and somewhat unexpected – we tend to think of snoRNAs as ubiquitous and “housekeeping”. The authors may want to highlight this observation in their discussion.

3. Related to this, the two cell types that express *jou* (enterocytes and ovarian nurse cells) are endoreplicating cells. Is *jou* required for their endoreplication? Does *jou* overexpression increase their ploidy? Do other endoreplicating cells express *jou*?

4. To begin to distinguish between possible effects on absorption and epithelial turnover, the authors could assess intestinal proliferation in both the *jou* loss- and gain-of function flies (which have shorter and longer lifespans, respectively).

5. Figure 2: what is the difference between panels C and D? Based on the main text and the figure legend, they both appear to show *jou* expression in nucleoli of midgut cells.

6. The transcriptional signature associated with *jou* loss includes many genes suggestive of changes in immunity/microbiota. The authors may want to discuss this, and perhaps explore whether the pro-longevity effects of *jou* are still present in germ-free flies.

7. As far as I can tell, all experiments were conducted using mated females flies. Does *jou* modulate lifespan in males too?

Reviewer #2 (Remarks to the Author):

Mellottée and colleagues present new findings related to a small nucleolar RNA (snoRNA), which they have named *jouvence*.

They report that *jouvence* is expressed in differentiated intestinal enterocytes (EC) and required to maintain normal lifespan in *Drosophila*. Moreover, the authors report that overexpression of *jouvence* in the intestine is sufficient to extend fly lifespan. To gain mechanistic insight the authors have performed gene expression analysis. Finally, the authors have identified putative a *jouvence* ortholog in mammals.

While some of the findings are potentially important, numerous technical and conceptual concerns dampen enthusiasm and should be addressed before publication. In addition, it is disappointing that there is little (if any) insight into the downstream mechanism(s).

Technical and conceptual concerns:

Fig 1; The 'rescue construct' contains what exactly? It seems that it contains more than just 'jouvence'. If so, how can the authors conclude that the short lifespan results from loss of jouvence?

Figure 3: There is no information provided regarding Myo1A-Gal4? Characterization and/or references should be provided.

3A-what exactly are the genotypes here? I am missing details in figure legend or results. In any case, the data are confusing and likely overinterpreted. In Fig. 1, the deletion line lived ~40 days. In Fig 3A, the 'deletion controls' lifespan is ~60 days. The addition of the jou transgene does further increase lifespan. It is, however, likely/possible that this reflects hybrid vigor.

Figs 3B, 3C-are not referred to in the results section. Again, I am missing details of genotypes. That said, the fact that the two 'controls' show such strikingly different phenotypes requires investigation/clarification.

The data in Fig. 3E is more compelling. It is very nice that the authors have made a new mex GS line. However, I am missing characterization of this line to demonstrate that it shows gut-specific inducibility.

Figure 4 could be supplementary data.

Fig 5 A, B. these data are likely confounded by genetic background differences. The progeny likely lives longer due to hybrid vigor. Indeed, please note the significant differences in lifespan between control lines. These data could be removed with no loss of information.

Fig. 5C. This is probably the most important and compelling data in the paper. How many times has this experiment been performed? Is the lifespan effect seen in both males and females? please show replicate experiments perhaps as a supplement figure. Please also show that in the authors' lab RU486 does not impact lifespan in control (+/mexGS) flies.

Figure 6 this data is potentially interesting but doesn't add much to the paper. It could be just mentioned in the discussion.

Figure 7. The gene expression data is potentially interesting. But, major technical problems exist. All the concerns regarding the lifespan data and lack of appropriate genetic controls holds true here also. It would greatly improve the paper if the authors would carry out this type of analysis in *jou>MexGS* flies with and without RU-mediated induction. Moreover, it would greatly improve the paper if the gene expression data could lead to a testable hypothesis to gain mechanistic insight.

At present, I am missing a clear interpretation of the results.

Reviewer #3 (Remarks to the Author):

Report on manuscript "Jouvence, a new small nucleolar RNA required in the gut extend lifespan in *Drosophila*", by L. Mellottée et al.

This manuscript reports the role of an H/ACA snoRNA PSI28S-1153, in the lifespan of *Drosophila*. This snoRNA is expressed in enterocytes and further appears to be conserved in mammals. RNA-Seq indicates that some genes are deregulated in the deletion mutants at the mRNA levels, providing as basis for the effects observed.

I found that the link between snoRNA and longevity quite interesting, and even more given the fact that this particular snoRNA appears conserved and expressed in a tissue-specific manner. The key experiment in the paper is in my mind the lifetime extension seen in overexpressing flies. This is a very important data because it would not be too surprising that lifetime is reduced in deletion mutants of snoRNAs, as they are believed to improve mRNA translation and in yeast snoRNA mutants show a reduced fitness in competition experiments. Some of the data presented in the paper seem however a bit preliminary and need however to be strengthened before the manuscript can be published.

Major points.

1-Lifespan extension. As said above, this is one of the key data of the paper. This phenomenon is seen in Fig1C, as well as in Figure 3A 3D 3E and Fig 5, where Jou is overexpressed from different drivers in different genetic backgrounds. However, the experiments in Fig 3 are all done from a single transgenic UAS-Jou. I would recommend to duplicate these results using a second, independent lines, perhaps where the transgene is integrated at another site in the fly genome. This might especially important for the comparison with snoRNA-2 and snoRNA-3 since these are inserted at a different site than Jou (VK27 vs. 68A4).

2-Longevity experiments. It is stated in the material and methods that each longevity experiment was tested 2 or 3 times, but it is not clear if what is shown is the result of a single experiment or the mean of several. Likewise, are the statistical tests performed on each experiment separately, or on the mean, etc... It would help to give the pValue of the test in the main panels, as well as showing the individual replicates of the key experiments (1C, 3A and 3D). In Figure 3B and 3C, is it known why the controls are so different from each other and what is the genotype of each control? An RU experiment should also be performed on a control strain to show it has no effect per se. Also, a longevity experiment from a wild type population would help interpret the experiment.

3-RNA expression. The expression of Jou in WT, Del, as well as the various overexpression strains, should be checked by Northern blot or RNase protection to provide a direct measurement of the RNA size and ensures it is correctly processed. Likewise, tissue-specific expression of Jou in *Drosophila* and mouse should be verified by Northern or RNase protection since these allow a direct and much better assessment of expression levels than in situ or PCR. More details should be given on how Jou is expressed, since most snoRNAs are not processed and assembled properly when inserted in exons. Are the human and mouse snoRNAs located in introns of host genes, and if yes, what are these genes and are they specifically expressed in the gut ?

Figure S2 and S3 are somehow of higher quality than Figure 2 and S4 (A, B, C). Why not put figS2A7 and S3 in the main manuscript?

4-Mechanism of action of Jou. This snoRNA is thought to catalyze modification of an rRNA base (28S-1153). Is this modification specifically present on ribosomes from the gut, as expected from the tissue-specific expression of Jou ? This would be quite interesting in itself and it would be easy to check using traditional RNA modification measurement techniques. It may also give some insights into the mechanism of action of Jou. Does the human/mouse ortholog of Jou target the same rRNA base ?

Another potential would be to do the overexpression experiment with a point mutant of Jou, in the pseudouridylation pocket.

Minor comment:

1-Abstract: "... indicating that the snoRNA-Jouvence might be involved in transcriptional control. This is really far-fetched given the data presented and this sentence should be removed.

2-introduction p3: snoRNAs in Archaea are called sRNA rather than snoRNAs. It is also incorrect to state that H/ACA snoRNAs are a "lesser known subtype". It should also be stated that snoRNAs can target the spliceosomal snRNAs (it is their main targets after rRNAs).

3-Results p4: some details should be given on the screen that allowed to isolate Jou.

4-It would be interesting to characterize expression of Jou during embryogenesis.

5-p7 to: "... cell types by combing in situ...". I suppose the authors mean "combining". Some words seem to be missing from the last sentence of the same paragraph.

6-Figure 4B and 4C: is the control the WT CS strain ? if not, it should be, or the WT CS levels should be indicated in the same graphs. Also, please try to keep the same colour code for each different genetic background, this will facilitate the understanding of the survival curve.

7-Starvation assays: I think that these results are very confusing and should be removed. In addition, they are not correctly referenced (Fig 6C1 instead of 9).

8-Discussion p14: "Preserving" should read "Preventing" ?

9-Discussion p15: the interpretation that Jou acts at the transcriptional level is far-fetched as there are many things that could disregulate gene expression indirectly, including suboptimal ribosome functioning.

10-Longevity experiment should be displayed using the same scale on the x-axis.

11-For the in situs in the main and supps panels, a schematic of the tissues displayed would be helpful for unfamiliar readers.

12-The transcriptomic analysis lacks precision: “catalytic activity” for instance, could encompass different activities from chromatin modifications to metabolic pathways while “cellular process” could apply to any cell component. The authors may want to re-process the data to check for functional pathway enrichment.

13-Figure 6: “sequence comparison” but not “homology”. Please indicate H/ACA boxes, and putative targeted sequence. What is the putative Jou substrate and does it match to the putative substrate of the supposedly homologous sno (PSI-1153) ?

14- Figure S1C: “controls” and “deletion” lanes may have been inverted.

Gif-sur-Yvette, June 1st, 2019

Jean-René Martin
Institut des Neurosciences Paris-Saclay (Neuro-PSI)
UMR-9197, CNRS/Université Paris Sud,
Team: Functional Brain Imaging and Behavior (FBIB)
1 Avenue de la Terrasse (Bat. 32/33)
91198, Gif-sur-Yvette, France
Tel: (33) 01.69.82.41.80 / Fax: (33) 01.69.82.34.47
e-mail: jean-rene.martin@inaf.cnrs-gif.fr

Manuscript number: NCOMMS-18-15686

jouvence, a new small nucleolar RNA required in the gut extends lifespan in *Drosophila*

Reviewers' comments:

Reviewer #1 (Remarks to the Author):

This manuscript describes the interesting observation that the levels of a small nucleolar RNA (*jou*) modulate lifespan from intestinal enterocytes in adult *Drosophila*. In a series of carefully controlled experiments, the authors show that *jou* is specifically expressed in enterocytes and ovarian cells. Flies lacking *jou* are viable but shorter-lived. One might argue that a shorter lifespan is only to be expected from losing a “housekeeping” function but, strikingly, *jou* overexpression exclusively in adult enterocytes is sufficient to increase lifespan substantially. The experiments provided are unusually well controlled; flies have been backcrossed for several generations to two different reference stocks, several overexpression lines have been used, a possible contribution of two neighbouring genes has been ruled out, two different enterocyte drivers have been used and *jou* expression levels have been carefully quantified in both loss- and a gain-of-function experiments.

Mechanistic exploration of how *jou* modulates lifespan includes transcriptional profiling of guts lacking *jou* relative to controls, and description of the effects of *jou* absence on desiccation/starvation resistance. Although such analyses are relatively superficial, the initial observation is novel and well documented. The authors also perform computational analyses that have revealed mammalian orthologues of *jou*. Their initial expression analyses indicate that at least one of them is expressed in the mouse intestine, raising the possibility that their *Drosophila* findings are more broadly significant.

We would like to thank this reviewer for his/her general appreciation of our study and highlighting our careful controls.

Specific comments

1. The authors need to mention Tiku et al 2016 (PMID: 28853436). Although their manuscript extends the findings described by Tiku et al in various ways (they show a role in flies rather than worms, they identify a specific cell population and a specific snoRNA involved), it is important to put these findings in the context of previous links between the nucleolus and lifespan. The authors may also want to discuss their findings in the context of known links between Myc/ribosome biogenesis and lifespan, and known correlations between reduced ribosomal protein content and increased lifespan in worms.

Response: Thanks for pointing this out.

1) We have now cited Tiku et al., page 21 and add the following paragraph in the Discussion: (Recently, in *C. elegans*, a link between the nucleolus size and lifespan involved in the determination of longevity).

2) Moreover, we have also discussed the putative role of Myc related to ribosome biogenesis in the Discussion (page 23) and add the following sentences (The second striking observation revealed by the RNA-Seq concerns the reported link between the Myc/ribosomes biogenesis and lifespan, Therefore, again here, our results argue rather that *jou* acts through new mechanisms or pathways to modulate the longevity).

2. The cell type specificity of *jou* expression is striking and somewhat unexpected – we tend to think of snoRNAs as ubiquitous and “housekeeping”. The authors may want to highlight this observation in their discussion.

Response: Good point. We now discuss this on page 24 (end of the first paragraph), and add the following sentence:

(..... Finally, although it is commonly considered that the snoRNAs might mainly be an ubiquitous and housekeeping machinery in all cells, obviously, as for the methylation of ribosomal RNA by NSUN5 [66], our results rather argues for a specific snoRNA machinery in certain tissues, as for instance here, in the epithelium of the gut).

3. Related to this, the two cell types that express *jou* (enterocytes and ovarian nurse cells) are endoreplicating cells. Is *jou* required for their endoreplication? Does *jou* overexpression increase their ploidy? Do other endoreplicating cells express *jou*?

Response: This was a very good and relevant observation from the reviewer. We have now carefully checked the endoreplication in the *jou*-deleted flies, as well as in various transgenic fly lines. In brief and as mentioned in the revised version, we find that:

- 1) Yes, *jou* is required for the endoreplication of enterocytes.
- 2) Yes, overexpression of *jou* increases the polyploidy.
- 3) No, not all the endoreplicating cells express *jou*.

More precisely, to assess if *jou* could be involved (directly or indirectly), in the endoreplication, we have analyzed the endoreplication in the enterocytes, by quantifying the DAPI-staining, which is proportional to the quantity of DNA (a method used and reported by previous authors, cited in the manuscript). Accordingly, we have also quantified, by RT-qPCR, the level of Cyclin E, one of the main genes involved in the cell cycle and endoreplication process. These results are presented in a new Figures (Figures 5E and F) (see pages 12 and 13 and in Discussion, page 21, first paragraph). In brief, *jou* is required for their endoreplication, while the overexpression of *jou* increases the polyploidy. However, not all the endoreplicative tissues expressed *jou*. Indeed, among these several tissues, only the nurse cells of the ovaries and the enterocytes of the gut express *jou*. Finally and interestingly, these endoreplication defects are rescued in transgenic fly lines, as well as it is increased when we overexpress *jou*.

4. To begin to distinguish between possible effects on absorption and epithelial turnover, the authors could assess intestinal proliferation in both the *jou* loss- and gain-of function flies (which have shorter and longer lifespans, respectively).

Response: Again, this was a very good and relevant critic from the reviewer, and then we have carefully analyzed the intestinal lesions, as proliferation, putative hyper-proliferation and dysplasia. In brief, we have analyzed several parameters and quantified several cell types.

1) Count the number of enterocytes directly by using DAPI-staining (new Figure 5A). We found that *jou*-deleted flies have more enterocytes (hyperplasia) than controls, while the overexpression of *jouvence* prevents this hyperplasia.

2) Count the number of ISCs using Delta-Gal4 line to label them (new Figure 5B), and found that the number of Delta+ cells are increased in *jou*-deleted flies.

3) Count the number of dividing cells using anti-PH3 immunostaining (new Figure 5C), and found that the number of dividing cells are increased in *jou*-deleted flies.

4) We have also used the EdU (an update version of the BrdU) which labels not only the cells in division, but also the endoreplicative cells (since EdU is incorporated into the DNA during the replication) (new Figure 5D), and found that the number of EdU+ cells are increased in *jou*-deleted flies, while the overexpression of *jouvence* prevents this over-proliferation.

Altogether, these results demonstrate that the hyperplasia observed in *jou*-deleted flies, both in young and old flies, is due to an over-proliferation of cells (ISCs). Accordingly, we have discussed these new results pages 20-21.

5. Figure 2: what is the difference between panels C and D? Based on the main text and the figure legend, they both appear to show *jou* expression in nucleoli of midgut cells.

Response: Yes, the reviewer is right. It has no difference between these 2 figures. It was just a repetition with a higher magnification between Figure 2C and D in order to better present the labelling. Then we have removed the Figure 2C and keep only the higher magnification. Then, former Figure 2D becomes A4, since on request of the reviewer-2, we have modified the whole Figure 2.

6. The transcriptional signature associated with *jou* loss includes many genes suggestive of changes in immunity/microbiota. The authors may want to discuss this, and perhaps explore whether the pro-longevity effects of *jou* are still present in germ-free flies.

Response: Although the microbiota effect is surely an interesting parameter *per se*, that could modulate the longevity of the flies, we consider that this highly specialized question is out of the focus of this study which describes the identification and characterization of a new snoRNA in relation to longevity. Therefore, we feel that this is outside the scope of our current study.

7. As far as I can tell, all experiments were conducted using mated females flies. Does *jou* modulate lifespan in males too?

Response: As suggested, we have added the Males Longevity (new Supplementary Figure S1F). Interestingly, the longevity in *jou*-deleted males is the inverse than in females (*jou*-deleted males live longer than control). However, such effect is not surprising, since a sexual dimorphism has already been described for several genes and parameters related to longevity, and so in many organisms. However, for the moment, we have not pursued or explored further this intriguing observation, because we consider that this represents a full study *per se*. Accordingly, a short paragraph has been added: first paragraph, page 6.

Reviewer #2 (Remarks to the Author):

Mellottée and colleagues present new findings related to a small nucleolar RNA (snoRNA), which they have named *jouvence*.

They report that *jouvence* is expressed in differentiated intestinal enterocytes (EC) and required to maintain normal lifespan in *Drosophila*. Moreover, the authors report that overexpression of *jouvence* in the intestine is sufficient to extend fly lifespan. To gain mechanistic insight the authors have performed gene expression analysis. Finally, the authors have identified putative a *jouvence* ortholog in mammals.

While some of the findings are potentially important, numerous technical and conceptual concerns dampen enthusiasm and should be addressed before publication. In addition, it is disappointing that there is little (if any) insight into the downstream mechanism(s).

Technical and conceptual concerns:

Fig 1; The 'rescue construct' contains what exactly? It seems that it contains more than just '*jouvence*'. If so, how can the authors conclude that the short lifespan results from loss of *jouvence*?

Response: as we have written and as presented in the Figure-1, the “rescue” lines corresponds to the red line in Figure-1, and contains the DNA genomic region of 1723 bp encompassing the 3 snoRNAs (sno-2, sno-3 and *jouvence*) as well as a short part of the 5'-region of the CG13334 and of the 3'-region of CG13333. This is detailed in the methods section. We agree that this construct alone (which contains the 3 snoRNAs) does not allow to discern which snoRNAs is responsible for the longevity phenotype (shorter or longer lifespan). However, in the aim to demonstrate the role of the snoRNA *jouvence*, and to discern the role of each snoRNA, we have generated several supplementary transgenic constructs (UAS-*jou*^{8M}, pJFRC-MUH-*jou*^{1M}, UAS-sno-2 and UAS-sno-3) in which we specifically target each single snoRNA in the epithelium of the gut, whose allow to precisely point-out which one is involved. Indeed, two different gut driver Gal4 lines (Myo1A and Mex) combined to 2 different UAS-*jouvence* constructs (*jou*^{8M} and pJFRC-MUH-*jou*^{1M}) lead to similar results (an increase of lifespan), whereas the targeted expression of the sno-2 or sno-3 does not rescue or increase lifespan. Therefore, altogether these results clearly demonstrate that *jouvence* is the snoRNA responsible for the longevity, and it is required in the enterocytes.

Figure 3: There is no information provided regarding Myo1A-Gal4? Characterization and/or references should be provided.

Response: This information and references appeared in the first paragraph of the page 7, and were included in the cited references 44 to 49 (see sentence: “The gut epithelium is composed of four cell-types: enterocytes (ECs), enteroblasts (EBs), entero-endocrine cells (EEs) and intestinal stems cells (ISCs) [44-49: now 41-44]”). However, in such context, I agree that the Myo1A-Gal4 was not enough precisely and directly referenced. Then, we have added some descriptive sentences and 2 references, citing Morgan, N.S., et al. J. Mol. Biol. 239, 347–356 [45] and Jiang et al., 2009 [46]: (added sentences, p. 7:To target and label the ECs, we used the Myo1A-Gal4 driver (NP1-Gal4), an enhancer trap in myosin 1A gene, encoding a gut specific brush border [45]. As described by Jiang et al., 2009 [46], Myo1A-Gal4 is strongly

expressed in all midgut ECs, while no expression was detected in ISCs, EBs, EEs, or visceral muscle.....).

3A-what exactly are the genotypes here? I am missing details in figure legend or results. In any case, the data are confusing and likely overinterpreted. In Fig. 1, the deletion line lived ~40 days. In Fig 3A, the 'deletion controls' lifespan is ~60 days. The addition of the jou transgene does further increase lifespan. It is, however, likely/possible that this reflects hybrid vigor.

Response: More precisely, the various genotypes are:

- 1) Del,Myo1A-Gal4 >UAS-jou^{8M},
- 2) Control-1= Deletion,Myo1A/Deletion (Myo1A-Gal4 in heterozygous)
- 3) Control-2 = Del;UAS-jou^{8M}/Del (UAS-jou^{8M} in heterozygous, without the driver line)

Concerning the difference between the original Deletion line reported in Figure-1B and the Del,Myo1A/Del (Myo1A-Gal4 in heterozygous) (Control-1), we have re-checked very carefully this difference and found that it is due to the insertional effect of the Myo1A-Gal4 itself (an enhancer-trap line) in the “deletion” background.

To do that, we have re-isolated from the parental line Del,Myo1A-Gal4/Del (Myo1A-Gal4 was permanently kept in heterozygous since it is not very well viable in homozygote) the Deletion itself (in homozygous: Del/Del) and re-quantify and re-compare the longevity of these various lines. These new results are presented in Supplementary Figure S4F, in which we can observe that the Deletion (in homozygous) recovered from the Del,Myo1A-Gal4 line lives as the original Deletion line presented in Figure-1B (median of about 30 days), while the Del,Myo1A-Gal4/Del still lives much longer (median of about 55 days). This indicates an insertional effect of the Myo1A-Gal4 line specifically in the Deletion genetic background, and so even visible in heterozygous. Nevertheless, this new finding does not impact our original interpretation, since the targeted expression of jouvence in the enterocytes increases lifespan.

Figs 3B, 3C-are not referred to in the results section. Again, I am missing details of genotypes. That said, the fact that the two 'controls' show such strikingly different phenotypes requires investigation/clarification.

Response: As explained just above (point-3A), the Control-1 corresponds to the Del,Myo1A/Del (Myo1A-Gal4 in heterozygous), and this quite long lifespan is due to the insertional effect of the enhancer-trap line Myo1A-Gal4. For the 2 other controls, which correspond to Del,sno-2/Del and Del,sno-3/Del, they show a quite short lifespan, because the UAS-sno-2 and UAS-sno-3 alone (without Gal4 driver) are obviously deleterious (or toxic) (as mentioned and discussed in the text).

The data in Fig. 3E is more compelling. It is very nice that the authors have made a new mex GS line. However, I am missing characterization of this line to demonstrate that it shows gut-specific inducibility.

Response: We now show that the Mex-GS>UAS-GFP is indeed expressed in the gut after feeding the flies with RU486, compared to non-fed flies. This result is now presented in Supplementary Figure S5F-G.

Figure 4 could be supplementary data.

Response: As requested, we have transferred this Figure in Suppl. Figures (now Figure S6)

Fig 5 A, B. these data are likely confounded by genetic background differences. The progeny likely lives longer due to hybrid vigor. Indeed, please note the significant differences in lifespan between control lines. These data could be removed with no loss of information.

Response: First, the Figure 5 is now becoming the Figure 4. As mentioned above in point No 3A, again here, obviously the Myo1A-Gal4 (an enhancer-trap line) has an effect *per se* on longevity. Though I agree that these results are not perfect, we thought that it is still relevant to present it because it clearly demonstrates that the overexpression of jouvence is sufficient to increase lifespan, even if one of the parental line show some side effect *per se*.

Additionally and noticeably, the Mex-Gal4 line (Control-1: Mex-Gal4/+, Figure 4B, formerly Figure 5B) does not present such difference between the two controls, indicating that this side effect is observed only with Myo1A-Gal4 line. Thus, very likely, it is not due to hybrid vigor, but to an insertional effect of the enhancer-trap P[Gal4] in the Myo1A gene.

Fig. 5C. This is probably the most important and compelling data in the paper. How many times has this experiment been performed? Is the lifespan effect seen in both males and females? please show replicate experiments perhaps as a supplement figure. Please also show that in the authors' lab RU486 does not impact lifespan in control (+/mexGS) flies.

Response: The experiment has been repeated a second time with the new construct (Mex-GS>pJFRC-MUH-jou^{1M}, with and without RU486) (new suppl. Figure S5B).

Moreover, the over-expression experiment using Mex-Gal4 has also been performed with the second jou transgenic construct (Mex>pJFRC-MUH-UAS-jou^{1M}) (new Suppl. Figure S5A).

Moreover, to rule-out that this effect could be a side effect due to the feeding of RU486, we have performed 3 independent controls (Mex-GS/+), (UAS-jou^{8M}/+), and (pJFRC-MUH-jou^{1M}/+) (Suppl. Figure S5-C,D,E respectively), comparing fed versus non-fed flies with RU486. In all three controls, the RU486 did not yield to any effect.

Figure 6 this data is potentially interesting but doesn't add much to the paper. It could be just mentioned in the discussion.

Response: As mentioned by the reviewer, I agree that these results are not the most important of the manuscript. Nevertheless, since the snoRNA are known to be generally well conserved through evolution, and so both structurally and functionally, and since the snoRNA jouvence is novel in Drosophila, we thought that it might be interesting to check if it is conserved through evolution, in a goal to enlarge the importance of this study (and not keep it strictly to Drosophila field). We therefore search and found its homologues, first in other Drosophila species and second, in mammals (mouse and human). Although I agree that we have not yet completely demonstrated the role of the jouvence orthologs in mammals, however, both the RT-qPCR as well as Northern Blot performed on gut tissue (as requested by the reviewer 3), demonstrate that they are expressed, and therefore very likely functional. We also agree that many further investigations will be required to decipher their role in mammals, including human, but we consider that this task is out of the focus of the current article. Thus, in this context, and since the reviewer 3 found these results "quite interesting", then in front of this dilemma, we decide to keep the Figure (the revised version which fulfils the other requests of the reviewer 3).

Figure 7. The gene expression data is potentially interesting. But, major technical problems exist. All the concerns regarding the lifespan data and lack of appropriate genetic controls holds true here also.

Response: We have performed a RNA-Seq comparing the Wild-Type Control and the mutant (deletion). The reviewer raises a good point, since the RNA-Seq has been performed on the deletion line, which encompasses the 3 snoRNAs. However, the independent validation of some deregulated genes, by RT-qPCR, for which the expression levels are rescued, both by the genomic-rescued line (rescue-1) and more importantly by the targeted expression of *jouvence* in the enterocytes (Del,Myo1A>jou) (presented in Figure 8) clearly demonstrate that the deregulation of genes is due to *jouvence* and not to the 2 other snoRNAs.

It would greatly improve the paper if the authors would carry out this type of analysis in *jou>MexGS* flies with and without RU-mediated induction. Moreover, it would greatly improve the paper if the gene expression data could lead to a testable hypothesis to gain mechanistic insight. At present, I am missing a clear interpretation of the results.

Response: As requested, we have performed a novel RNA-Seq on the flies conditionally over-expressing *jouvence* only in adulthood, and these new results are now presented (new Figure 9 and Suppl. Figures). Remarkably, only eight genes are deregulated, noticeably a Heat-Shock Protein gene and the Cytochrome P450-6a8 gene, two genes that have already been implicated in lifespan determination as well as in stress resistance (two phenotypes that are affected by *jouvence* and that we have reported here). In brief, 2 hypothesis could be formulated:

1) for the overexpression of *jouvence*: these two genes, and more particularly the Hsp-70Bb, a chaperon protein which stabilizes many other proteins, are in accordance with the fact that stabilizing the proteins helps to optimize the longevity (this is also partly in agreement with the lack of function of *jouvence* in which the ribosomes are not pseudouridylated, which might yield to perturbation in protein synthesis).

2) for the deletion and rescued of *jouvence*: as revealed by the RNA-seq, about 600 hundred genes are deregulated (~300-up and ~300-down), revealing that the lack of *jouvence* has more dramatic effect on the level of mRNA of several genes than its overexpression. Nevertheless, the rescued of the mRNA levels of few genes (Figure 8) confirms the role of *jouvence* in this deregulation.

The drastic difference in the number of genes deregulation observed between the lack of function of *jouvence* and its over-expression is not an exceptional case (the lack of function of a gene could have very different consequences than its overexpression in a Wild-Type condition).

As we have demonstrated, *jouvence* is required for the pseudouridylation of ribosomal RNA, a function that might likely stabilize the ribosome and consequently improve the protein synthesis/turnover (a hypothesis that remain to be tested). Interestingly, the ribosome stabilization, which is important in the determination of the protein levels, have been clearly demonstrated to play a role in lifespan determination in *C. elegans* (ref. 66 and within).

Based on these results, we hypothesize that the main functional role of *jouvence* seems to reside in his role in ribosomes stabilization, through the pseudouridylation. Such ribosome stabilization/optimization could be tested, as for example, by performing a ribosome profiling study (a hypothesis that remains to be tested, but which is out of focus of this study).

We have now clearly formulated such hypothesis in the discussion (end of the first paragraph, page 23: (... Therefore, this *jou*-independent Myc regulation rather leads to hypothesis that *jouvence* may act, through the pseudouridylation, on ribosomes stability and/or optimize the ribosomes function (which argues for the concept of “specialized” ribosomes) [66] instead of acting directly on ribosome synthesis.)

Reviewer #3 (Remarks to the Author):

Report on manuscript "Jouvence, a new small nucleolar RNA required in the gut extend lifespan in *Drosophila*", by L. Mellottée et al.

This manuscript reports the role of an H/ACA snoRNA PSI28S-1153, in the lifespan of *Drosophila*. This snoRNA is expressed in enterocytes and further appears to be conserved in mammals. RNA-Seq indicates that some genes are deregulated in the deletion mutants at the mRNA levels, providing as basis for the effects observed.

I found that the link between snoRNA and longevity quite interesting, and even more given the fact that this particular snoRNA appears conserved and expressed in a tissue-specific manner. The key experiment in the paper is in my mind the lifetime extension seen in overexpressing flies. This is a very important data because it would not be too surprising that lifetime is reduced in deletion mutants of snoRNAs, as they are believed to improve mRNA translation and in yeast snoRNA mutants show a reduced fitness in competition experiments. Some of the data presented in the paper seem however a bit preliminary and need however to be strengthened before the manuscript can be published.

We would like to thank this reviewer for his/her general appreciation of our study and highlighting the fact that this particular snoRNA appears conserved and expressed in a tissue-specific manner.

Major points.

1-Lifespan extension. As said above, this is one of the key data of the paper. This phenomenon is seen in Fig1C, as well as in Figure 3A 3D 3E and Fig 5, where Jou is overexpressed from different drivers in different genetic backgrounds. However, the experiments in Fig 3 are all done from a single transgenic UAS-Jou. I would recommend to duplicate these results using a second, independent lines, perhaps where the transgene is integrated at another site in the fly genome. This might especially important for the comparison with snoRNA-2 and snoRNA-3 since these are inserted at a different site than Jou (VK27 vs. 68A4).

Response: This request partly overlaps with a point of the referee No 1. As requested, we have added a second independent replicate of the longevity experiment (Del,Myo1A>jou^{8M} presented in Suppl. Figures S4A) and our new findings support our original data indicating a rescued and even an extension of lifespan upon the targeted expression of jou in enterocytes. Moreover, we have also generated a second construct containing the snoRNA-jouvence in the pJFRC-MUH vector, and generated a new transgenic line (named: pJFRC-MUH-jou^{1M}) that we have directed the insertion on the chromosome 3, into attP2 site located at position 68A4 on the third chromosome. Then we have repeated the longevity experiments with all the gut Gal4 driver lines previously used as well as with Mex-GS. All of these new results are presented in the Suppl. Figures (S4 and S5). In brief, all of these combinations (Myo1A or Mex or Mex-GS >pJFRC-MUH-jou^{1M}) lead to an increase of lifespan similar to the UAS-jou^{8M} construct, both in the deletion genetic background (rescue), or in overexpression, thus confirming that jou is responsible of the longevity phenotype, and ruling-out putative insertional side effects.

2-Longevity experiments. It is stated in the material and methods that each longevity experiment was tested 2 or 3 times, but it is not clear if what is shown is the result of a single experiment or the mean of several. Likewise, are the statistical tests performed on each

experiment separately, or on the mean, etc... It would help to give the pValue of the test in the main panels, as well as showing the individual replicates of the key experiments (1C, 3A and 3D). In Figure 3B and 3C, is it known why the controls are so different from each other and what is the genotype of each control? An RU experiment should also be performed on a control strain to show it has no effect per se. Also, a longevity experiment from a wild type population would help interpret the experiment.

Response:

1) The presented longevity experiments were the results (raw data) from a single experiment. However, as formerly mentioned, some experiments were already done twice. In other words, the showed experiments were not the mean of several experiments but are the raw data of a single experiment. Then, as requested, we have added in the Suppl. Figures, the second replicate for the genomic-rescue-1 line (Suppl. Figure S1D) and Del,Myo1A>jou^{8M} (Suppl. Figure S4A).

2) The pValue (or rather its corresponding symbol: ex.: *** p<0,0001) has been added in the Figures, while the detailed parameters of all longevity experiments are shown in Suppl. Table S1). The replicates of key experiments are now presented in Suppl. Figures.

3) The genotype of each controls have now been more precisely written. Controls (Control-2 = Del;UAS-sno-2 and Del;UAS-sno-3 in Figures 3B and 3C respectively) are different from each other because they are obviously deleterious (toxic) in the deletion genetic background. I remind that the genetic background of all the lines have been thoroughly homogenized (Cantonized).

4) We have added 4 different controls for RU486 (Suppl. Figure S4E and Suppl. Figures S5C-D-E). In brief, none of them give a phenotype, meaning that RU486 *per se* do not have an effect on longevity. These results reinforce our conclusion that the observed effect with the Mex-GS are clearly due to the expression of jouvence, and even it is sufficient only in adulthood.

5) In each experiment, the appropriated Wild-Type controls have been performed simultaneously and strictly in parallel. They are presented for each experiment.

3-RNA expression. The expression of Jou in WT, Del, as well as the various overexpression strains, should be checked by (a) Northern blot or RNase protection to provide a direct measurement of the RNA size and ensures it is correctly processed. Likewise, tissue-specific expression of Jou in Drosophila and mouse should be verified by Northern or RNase protection since these allow a direct and much better assessment of expression levels than in situ or PCR. More details should be given on how Jou is expressed, since most snoRNAs are not processed and assembled properly when inserted in exons. (b) Are the human and mouse snoRNAs located in introns of host genes, and if yes, what are these genes and are they specifically expressed in the gut?

Response:

a) Northern Blot: We have performed Northern Blots in Drosophila on various transgenic lines (presented in Figures 6A), which show that the snoRNA-jouvence is detected at 148 nucleotides as expected. Northern Blots have also been performed in mouse, and we detect, as expected, the snoRNA-15 at 129 nucleotides, and the snoRNA-18 at 122 nucleotides (presented in Suppl. Figures S10B), demonstrating that these snoRNAs are expressed and correctly processed *in-vivo*.

b) To get some more details about how Jou is expressed, since most snoRNAs are not processed and assembled properly when inserted in exons, and to respond to the question: Are the human and mouse snoRNAs located in introns of host genes:

We have added the genomic map of the locus containing the snoRNA-jouvence in human, mouse-15 and mouse-18 (Suppl. Figure S11). In all of them, the snoRNA-jouvence is located in a long intron of a gene. We have also shown by RT-qPCR performed on total-RNA from the gut, that each of these genes are expressed in the gut (see Suppl. Figure S12). Remark that for the RT-qPCR, we have use primers located in two adjacent exons, over an intron, then confirming that we strictly amplified a cDNA (mRNA).

Figure S2 and S3 are somehow of higher quality than Figure 2 and S4 (A, B, C). Why not put figS2A7 and S3 in the main manuscript?

Response: Yes, this is a good suggestion. We have re-shaped the Figure 2 as follows: we have transferred the Figure S7A7 and A8 in the Figure 2 (now Figure D1 and D2), as well as Figure S3 in the Figure 2 (now Figure 2C1-C2-C3)

4-Mechanism of action of Jou. This snoRNA is thought to catalyze modification of an rRNA base (28S-1153). Is this modification specifically present on ribosomes from the gut, as expected from the tissue-specific expression of Jou ? This would be quite interesting in itself and it would be easy to check using traditional RNA modification measurement techniques. It may also give some insights into the mechanism of action of Jou. Does the human/mouse ortholog of Jou target the same rRNA base ?

Response:

1) This is a good and relevant suggestion. We have performed pseudouridylation experiments. As expected, we show that jouvence pseudouridylates the 28-rRNA at position 1153, as well as the 18S-rRNA at position 1425 (see new Figure 6B-C). Interestingly, the pseudouridylation is rescued in the genomic-rescue line (Del+rescue-1) as well as with targeted specific expression of jou in the gut (Del,Myo1A>jou^{8M}, and Del,Mex>jou^{8M}), demonstrating that this is jouvence that performs the pseudouridylation (and not potentially the 2 other snoRNAs).

2) Does the human/mouse ortholog of Jou target the same rRNA base ?

We could not directly respond to this question right now. Just by homology search, we did not find direct target, but due to the high complexity of the 3D structure of the rRNA, this “homology” approach is likely not stringent enough. Consequently, we could not exclude that it does pseudouridylation on other base of rRNA. The clear and definitive response should come from the pseudouridylation experiment itself, an experiment that is difficult be performed directly right now since we do not have a null mutant as comparison.

Another potential would be to do the overexpression experiment with a point mutant of Jou, in the pseudouridylation pocket.

Response: This is a good suggestion, but this will be a long term experiment, because we will need to generate a construct containing the mutation, then the transgenic animal and thereafter the longevity experiments. Thus, in the current context, since we have already clearly demonstrated that the pseudouridylation is performed by jouvence, we consider that this is not essential to perform such supplementary experiment.

Minor comment:

1-Abstract: "... indicating that the snoRNA-Jouvence might be involved in transcriptional control. This is really far-fetched given the data presented and this sentence should be removed.

Response: It has been removed.

2-introduction p3: snoRNAs in Archaea are called sRNA rather than snoRNAs. It is also incorrect to state that H/ACA snoRNAs are a "lesser known subtype". It should also be stated that snoRNAs can target the spliceosomal snRNAs (it is their main targets after rRNAs).

Response: These modifications have been done (see second paragraph page 3).

3-Results p4: some details should be given on the screen that allowed to isolate Jou.

Response: We have now described (page 4) more precisely the origin of the enhancer-trap P[Gal4]4C line and its genomic locus, in which jouvence is localized. (..... In a study to characterize putative genes involved in locomotor activity Here, in a step further, we molecularly characterize the P[Gal4]4C line, as well as we study it in the context of aging and particularly in aging effects on sensory-motor performance.....).

4- It would be interesting to characterize expression of Jou during embryogenesis.

Response: To characterize the expression of jou during embryogenesis and other developmental stages, we have performed RT-qPCR (results are presented in Suppl. Figures S13). We could see that jou is expressed (although faintly) at all development stages (including pupal stages), but with a stronger expression at the white pupal stage. These results are in accordance with the "modENCODE Temporal Expression Data" reported in FlyBase.

5-p7 to: "... cell types by combing in situ...". I suppose the authors mean "combining". Some words seem to be missing from the last sentence of the same paragraph.

Response: The mistakes have been corrected.

6-Figure 4B and 4C: is the control the WT CS strain ? if not, it should be, or the WT CS levels should be indicated in the same graphs. Also, please try to keep the same colour code for each different genetic background, this will facilitate the understanding of the survival curve.

Response:

a) Yes, the controls are the WT-CS.

b) The color code has been standardized in all graphs.

7-Starvation assays: I think that these results are very confusing and should be removed. In addition, they are not correctly referenced (Fig 6C1 instead of 9).

Response: I agree that these results are not the most important of the study. However, as mentioned by other reviewers, they contribute to the characterization of the general and organismal phenotype. Then, for this reason, we decide to keep these "secondary" results but we transfer all of the stress resistance tests in Suppl. Figure S8.

8-Discussion p14: "Preserving" should read "Preventing" ?

Response: Corrected !

9-Discussion p15: the interpretation that Jou acts at the transcriptional level is far-fetched as there are many things that could disregulate gene expression indirectly, including suboptimal ribosome functioning.

Response: Yes, the reviewer is right. This has been modified as follow (see Discussion, page 24, first paragraph). (... Therefore, the snoRNA-jouvence might likely act either in the maturation and/or stabilisation of the rRNA, although an involvement in the regulation of the transcription or even potentially in the chromatin structure, as already reported for some other snoRNAs [25] could not at this stage of our study, be excluded.)

10-Longevity experiment should be displayed using the same scale on the x-axis.

Response: We have reformatted and standardized all the Longevity graphs.

11-For the in situs in the main and supps panels, a schematic of the tissues displayed would be helpful for unfamiliar readers.

Response: A schematic drawing of the gut has been added in Figure 2E.

12-The transcriptomic analysis lacks precision: "catalytic activity" for instance, could encompass different activities from chromatin modifications to metabolic pathways while "cellular process" could apply to any cell component. The authors may want to re-process the data to check for functional pathway enrichment.

Response: We have re-analyzed all the RNA-Seq data (control versus deletion) (Novogene, Co). More precisely, we have performed a KEGG analysis, which give much more precise information about all the genes (and the number of genes) that are affected in each specific pathway. All of these new results are now available in Suppl. Information (see Suppl. Tables S3 to S11). A Gene Ontology (GO) analysis has also been performed and showed.

13-Figure 6: "sequence comparison" but not "homology". Please indicate H/ACA boxes, and putative targeted sequence. What is the putative Jou substrate and does it match to the putative substrate of the supposedly homologous sno (PSI-1153) ?

Response:

1) We have replaced the "homology" by "sequence comparison" and indicate the two boxes (H and ACA) in each orthologue sequence.

2) We have demonstrated, in Drosophila, that jouvence pseudouridylates the rRNA (28S and 18S). For the mammalian orthologues, as mentioned in the response to point No 4 just above, we have not directly identified the substrate by homology search. This will undoubtedly require further experimental analysis to confirm or infirm such putative pseudouridylation.

14- Figure S1C: "controls" and "deletion" lanes may have been inverted

Response: right, it was a mistake (sorry !)

Reviewers' comments:

Reviewer #1 (Remarks to the Author):

My questions/concerns have been addressed and the manuscript is improved, so I am happy to recommend its publication.

Reviewer #2 (Remarks to the Author):

I appreciate the time and effort that the authors have spent on this paper and project.

However, my major concerns from the previous review have not been addressed:

Major concern remains:

Lack of significant conceptual advance:

It is disappointing that there is little (if any) insight into the downstream mechanism(s).

Technical concern: I am missing characterization of the Mex-GS line to demonstrate that it shows gut-specific

inducibility. The authors now show that it can express in gut. But, they do not show that it is gut-specific, i.e. not expressed in brain, muscle, etc...

In the rebuttal letter, in response to the request for repeat experiments of the *jou* overexpression with *mexGS*, the authors state: "The experiment has been repeated a second time with the new construct (*Mex-GS*>*pJFRC-MUH-jou1M*, with and without *RU486*) (new suppl. Figure S5B)."

However, on page 10 of the manuscript the authors state: "We also quantify the level of the snoRNA on dissected gut (Suppl. Figure S5B) in the two gut-driver lines. "

so, there seems to be some error when referencing this figure and/or I am confused.

Reviewer #4 (Remarks to the Author):

Report on manuscript NCOMMS-18-15686, entitled “Jouvence, a new small nucleolar RNA required in the gut extends lifespan in *Drosophila*”

In this paper, Soulé et al. identify Jouvence, a snoRNA which expression increases lifespan of *Drosophila*. Jouvence expression is restricted to enterocytes and ovaries nurse cells, and correlates with decreased hyperplasia of the enterocytes from the gut epithelium.

In this revised version, the authors use various constructs to solidify their first affirmation. It appears that lack of jouvence decreases lifespan in females (which might be expected from a snoRNA as sub-optimal ribosome maturation would affect translation). More strikingly, jouvence overexpression in a wild type background increases lifespan. The authors used various strategies to maipulate jouvence expression (deletion mutants, transgenic constructs, inducible transgenes), with numerous controls, redundant constructs and experiments to solidify this point – including testing two commonly used *Drosophila* strains. They also provide extensive details about the experimental conditions and control considered (including back-crosses to avoid “hybrid vigor”), fully responding to the referees’ previous criticisms. In addition, the authors provide files with quantitative data describing the longevity experiments as well as systematic statistical analysis. Finally, expression levels and accurate processing of the various jouvence constructs were verified by qPCR, Northern blot and FISH. Now, the impact of jouvence on lifespan holds true for females only, since the same deletion increased lifespan in male flies, so that only the role of jouvence in females was investigated in this paper.

As proposed by a referee, the authors also tested if *jouvence* would improve resistance to two stresses: desiccation or starvation. While *jouvence* might strengthen resistance to desiccation (especially in old flies), the response to starvation was less obvious.

Interestingly, *jouvence* expression is restricted to the enterocytes of the gut epithelium, and nurse cells from ovaries, two cell types which undergo endoreplication. This analysis was performed by co-staining for *jouvence* and markers specific for the different gut cell types. As expected for a snoRNA, *jouvence* is detected in the nucleolus of the enterocytes. To verify the function of *jouvence* as an H/ACA snoRNA, the revised version now includes experiments showing site-specific pseudo-uridylation of 20s and 18s RNA in control or rescued animals, but not in animals with a deletion encompassing *jouvence* encoding-gene. This experiment strongly supports that *jouvence* directs rRNA modification by assembling as a H/ACA snoRNP. Finally, the author provide several data suggesting that *jouvence* homologs might also exist in mammals, with a restricted expression pattern including gut.

Further experiments show an inverse correlation with the number of enterocytes in the gut of flies and *jouvence* expression. Indeed, the authors show that the number of enterocytes per unit of epithelium surface increases with age in wild type flies, as expected. Deletion of the *jouvence* locus further increased the number of enterocytes in young and old flies, while transgenic *jouvence* expression decreased the number of enterocytes in old flies. Again this effect was observed in the deletion and the wild type background, taking into consideration several genotypes to modulate *jouvence* expression (deletion, alone, with a rescue transgene or inducible transgene under the control of two different drivers...). This hyperplasia likely results from an increased proliferation of the intestinal stem cells (ISCs) and differentiation into enterocytes since the authors observed more cell division, more stem cells and less endoreplicative cells (a marker of differentiation) in the deleted strain. From these observations, the authors conclude that *jouvence* affects lifespan by lowering enterocytes division. Increased number of gut cells, including enterocytes and ISC, with mis-differentiation is considered a hallmark of aging in *Drosophila*. This phenomenon called “dysplasia”, is linked with a reduced

regenerative capacity of the epithelium, a loss of barrier integrity and increased susceptibility to dysbiosis – ultimately reducing lifespan. While interactions between gut microbia / intestinal epithelium homeostasis and lifespan have been well documented, it is not clear to me if the observed decreased gut turnover would explain per se the effect of jouvence on lifespan. This point is eluded in the paper. Does jouvence delays epithelium dysfunction in old flies (it seems to me that permeability tests with “smurf” flies could be easily performed) ? Is the microbiota involved in jouvence effect on lifespan (there again, I feel that this question could be addressed quite simply by treating flies with antibiotics to check if, in the absence of microbiota, jouvence still increases lifespan)?

The authors provide transcriptomic analysis of RNA expression from gut epithelium, comparing flies with deletion for the jouvence locus and wt. Yet, the data fail to point to a mechanistic insight and probably mostly reflect a variation in the tissue composition. In addition, the authors provide an analysis of RNA expression in gut epithelium from wt flies vs. jouvence overexpression. This analysis uncovers only a few genes. Could that be because jouvence induction was not long enough (I did not find indication for the time of RU treatment)?

In conclusion, this new version of Soulé et al. identifies jouvence as a snoRNA expressed in nurse cells and gut epithelium. Jouvence expression reduces enterocytes proliferation and increases lifespan in female flies. This is a novel, interesting finding which raises several questions, including the molecular mechanisms underlying this effect.

Minor comments:

Figure S3 is nice and could be inserted in the main figures

Fig 7 :RNA seq data: please annotate figures so that it is immediately clear if analysis of jou in wt or overexpression

Keep figure format from figure 7 to 9

Figure 8 : as supplementary

S9: high degree of identity” instead of “homology”

S10: mis-labelling of snoRNA expression from mouse: keep the name “sno -15 “ or “sno-18”

(not sno-1 or -2) while “sno-18” should be removed from the U6 hybridization.

S12: The qPCR experiment does not allow to evaluate the number of copies detected for each mRNA. Why would the author claim that Ctnnd2 and EPC1 are “faintly” expressed?

Gif-sur-Yvette, Septembre 18th, 2019

Jean-René Martin
Institut des Neurosciences Paris-Saclay (Neuro-PSI)
UMR-9197, CNRS/Université Paris Sud,
Team: Functional Brain Imaging and Behavior (FBIB)
1 Avenue de la Terrasse (Bat. 32/33)
91198, Gif-sur-Yvette, France
Tel: (33) 01.69.82.41.80 / Fax: (33) 01.69.82.34.47
e-mail: jean-rene.martin@inaf.cnrs-gif.fr

Manuscript number: NCOMMS-18-15686B
jouvence, a new small nucleolar RNA required in the gut extends lifespan in *Drosophila*

Reviewers' comments:

Reviewer #1 (Remarks to the Author):

My questions/concerns have been addressed and the manuscript is improved, so I am happy to recommend its publication.

We would like to thank this reviewer for his/her general appreciation of our study.

Reviewer #2 (Remarks to the Author):

I appreciate the time and effort that the authors have spent on this paper and project. However, my major concerns from the previous review have not been addressed:

Major concern remains:

Lack of significant conceptual advance:

It is disappointing that there is little (if any) insight into the downstream mechanism(s).

We have now added two new results (Figure 8G and H) showing that when we rescue the mRNA level of some deregulated genes, specifically in the epithelium of the gut, its nicely leads to an increase (rescue) lifespan, and so, when we express them only in adulthood, using the Gene-Switch system. More precisely, we have selected two of them: one up-regulated gene (*ninaD*) and one down-regulated gene (*CG6296*). First, in the aim to decrease and restore the mRNA level of *ninaD*, we show that the targeted expression of a *ninaD*-RNAi specifically in the gut, and more precisely only in adulthood (*Del,Mex-GS>UAS-ninaD-RNAi* with and without RU486), is sufficient to importantly decrease and restore the mRNA level (quantified by RT-qPCR) (Figure 8G-inset), and consequently increases (rescue) lifespan (Figure 8G). Second but inversely, for the *CG6296* (one of the most down-regulated genes), to restore the level of its transcript, we constructed an UAS-cDNA and a transgenic fly-line. The targeted expression of the cDNA of *CG6296* (*Del,Mex-GS>pJFRC-MUH-cDNA-CG6396* with and without RU486), specifically in the gut and only in adulthood, restores the level of the mRNA (quantified by RT-qPCR) (Figure 8H-inset), and also leads to an important increase of lifespan. These new results clearly indicate that the effect of the snoRNA *jouvence* is conveyed by the

different deregulated genes (revealed by the RNA-Seq) and therefore represents a mechanistic insight (genetic and molecular link) on how jouvence molecularly relays its effect on longevity. Altogether, we think that these new results highlight a genetic and molecular link between the effect of the snoRNA on ribosomal RNA, the downstream deregulated genes and lifespan.

For reminder, in brief, we have characterized and shown, **for the first time**:

- 1) Identify and molecularly characterize a new snoRNA (jouvence) of H/ACA box
- 2) Show that jouvence is expressed and required in the epithelium of the gut
- 3) The overexpression of jouvence prevents the hyperplasia/dysplasia of the gut in old flies
- 4) snoRNA-jouvence (a new H/ACA box) pseudouridylates the ribosomal RNA.
- 5) Transcriptomic analysis have revealed that several genes are deregulated (up or down)
- 6) The rescue of the mRNA level of two of them (ninaD and/or CG6296), in the deletion background, is sufficient to increase (and rescue) lifespan.
- 7) jouvence is conserved through evolution and we have shown that it is expressed in mouse and human (and then it might be functional).

Therefore, to my point of view, I think that all of these new and original results represent significant conceptual advances in longevity field.

Technical concern: I am missing characterization of the Mex-GS line to demonstrate that it shows gut-specific inducibility. The authors now show that it can express in gut. But, they do not show that it is gut-specific, i.e. not expressed in brain, muscle, etc..

We have carefully checked **again** the expression pattern of the Mex-GS line induced with RU486 at the dose of 25 µg/ml (the dose that we have used in our study), and we confirm that we don't detect any expression in other tissues (brain, muscles, etc.). We did not previously present these negative results, as it is usual for any other studies, and also because the article had already numerous figures. Nevertheless, to fill this new request, we have now added, in Suppl. Figures S14, an image showing simultaneously, a part of the abdomen (gut), the thorax (muscle) and the head (brain) on which we can see that the Mex-GS driving the UAS-GFP-nls after RU486 induction is expressed specifically and only in the gut.

In the rebuttal letter, in response to the request for repeat experiments of the jou overexpression with mexGS, the authors state: "The experiment has been repeated a second time with the new construct (Mex-GS>pJFRC-MUH-jou1M, with and without RU486) (new suppl. Figure S5B)." However, on page 10 of the manuscript the authors state: "We also quantify the level of the snoRNA on dissected gut (Suppl. Figure S5B) in the two gut-driver lines. " so, there seems to be some error when referencing this figure and/or I am confused.

Yes, it was a mistake from me. This has to refer to Suppl. Figure S6 (and not to Figure S5B). Sorry for this mistake and Thanks to the reviewer to have identify it. I have corrected it in page 10 of the manuscript.

Reviewer #4 (Remarks to the Author):

Report on manuscript NCOMMS-18-15686, entitled "Jouvence, a new small nucleolar RNA required in the gut extends lifespan in *Drosophila*"

In this paper, Soulé et al. identify Jouvence, a snoRNA which expression increases lifespan of *Drosophila*. Jouvence expression is restricted to enterocytes and ovaries nurse cells, and correlates with decreased hyperplasia of the enterocytes from the gut epithelium. In this revised version, the authors use various constructs to solidify their first affirmation. It appears that lack of Jouvence decreases lifespan in females (which might be expected from a snoRNA as sub-optimal ribosome maturation would affect translation). More strikingly, Jouvence overexpression in a wild type background increases lifespan. The authors used various strategies to manipulate Jouvence expression (deletion mutants, transgenic constructs, inducible transgenes), with numerous controls, redundant constructs and experiments to solidify this point – including testing two commonly used *Drosophila* strains. They also provide extensive details about the experimental conditions and control considered (including back-crosses to avoid “hybrid vigor”), fully responding to the referees’ previous criticisms. In addition, the authors provide files with quantitative data describing the longevity experiments as well as systematic statistical analysis. Finally, expression levels and accurate processing of the various Jouvence constructs were verified by qPCR, Northern blot and FISH. Now, the impact of Jouvence on lifespan holds true for females only, since the same deletion increased lifespan in male flies, so that only the role of Jouvence in females was investigated in this paper.

As proposed by a referee, the authors also tested if Jouvence would improve resistance to two stresses: desiccation or starvation. While Jouvence might strengthen resistance to desiccation (especially in old flies), the response to starvation was less obvious. Interestingly, Jouvence expression is restricted to the enterocytes of the gut epithelium, and nurse cells from ovaries, two cell types which undergo endoreplication. This analysis was performed by co-staining for Jouvence and markers specific for the different gut cell types. As expected for a snoRNA, Jouvence is detected in the nucleolus of the enterocytes. To verify the function of Jouvence as an H/ACA snoRNA, the revised version now includes experiments showing site-specific pseudo-uridylation of 20s and 18s RNA in control or rescued animals, but not in animals with a deletion encompassing Jouvence encoding-gene. This experiment strongly supports that Jouvence directs rRNA modification by assembling as a H/ACA snoRNP. Finally, the author provide several data suggesting that Jouvence homologs might also exist in mammals, with a restricted expression pattern including gut.

Further experiments show an inverse correlation with the number of enterocytes in the gut of flies and Jouvence expression. Indeed, the authors show that the number of enterocytes per unit of epithelium surface increases with age in wild type flies, as expected. Deletion of the Jouvence locus further increased the number of enterocytes in young and old flies, while transgenic Jouvence expression decreased the number of enterocytes in old flies. Again this effect was observed in the deletion and the wild type background, taking into consideration several genotypes to modulate Jouvence expression (deletion, alone, with a rescue transgene or inducible transgene under the control of two different drivers...). This hyperplasia likely results from an increased proliferation of the intestinal stem cells (ISCs) and differentiation into enterocytes since the authors observed more cell division, more stem cells and less endoreplicative cells (a marker of differentiation) in the deleted strain. From these observations, the authors conclude that Jouvence affects lifespan by lowering enterocytes division. Increased number of gut cells, including enterocytes and ISC, with mis-differentiation is considered a hallmark of aging in *Drosophila*. This phenomenon called “dysplasia”, is linked with a reduced regenerative capacity of the epithelium, a loss of barrier integrity and increased susceptibility

to dysbiosis – ultimately reducing lifespan. While interactions between gut microbiota / intestinal epithelium homeostasis and lifespan have been well documented, it is not clear to me if the observed decreased gut turnover would explain per se the effect of jouvence on lifespan. This point is eluded in the paper. Does jouvence delays epithelium dysfunction in old flies (it seems to me that permeability tests with “smurf” flies could be easily performed) ? Is the microbiota involved in jouvence effect on lifespan (there again, I feel that this question could be addressed quite simply by treating flies with antibiotics to check if, in the absence of microbiota, jouvence still increases lifespan)?

We have not provided results about the microbiota effect on these flies, because we think that it is out of the scope of this study. It will surely be interesting, but it undoubtedly represents a full study *per se*. Notably, to lead to real and conclusive results about microbiota, it will require many precise and dedicated experiments, which as we mentioned above, represents a full study *per se*.

I think that the message of the current study is already very dense and I very seriously think that adding other new results will not make the message better and more understandable, but will rather only participate to scramble the current message. Too much is not necessarily better and will lead the message fuzzy because all could not be joined in a single and simple interpretation. As requested, we have checked, at least as best as it can be done up to now, the “intestinal integrity” by counting the “smurf” flies. However, we did not detect any significant difference in the number of “smurf” flies, between Control-WT and Deletion flies (even we did not detect almost any smurf flies in these various groups of flies). For this reason, we did not add any Histogram in the result section.

Nevertheless, we have added, in the Discussion section, page 22, the following sentence: “Finally, we also check for the intestinal integrity of the flies, the so called “smurf” flies [11,16,60], at different ages (7 - 20 and 40 day-old flies), but we did not notice any significant difference between the Control-WT and the deletion flies (since in all of them, we observe almost no smurf flies, for about 100 flies for each group).”

I just want to emphasis here that according to the literature [references: 11,16,60], the number of “smurf flies” (and percentage) seems to be quite variable in the various experiments/laboratories, and even sometime it is very low (just few percent: example - reference 16). This suggests that likely many other factors could be involved, as perhaps the food medium. For example, here in our study, our food medium does not contain any sugar.

The authors provide transcriptomic analysis of RNA expression from gut epithelium, comparing flies with deletion for the jouvence locus and wt. Yet, the data fail to point to a mechanistic insight and probably mostly reflect a variation in the tissue composition. In addition, the authors provide an analysis of RNA expression in gut epithelium from wt flies vs. jouvence overexpression. This analysis uncovers only a few genes. Could that be because jouvence induction was not long enough (I did not find indication for the time of RU treatment)?

In this specific case (overexpression: Mex-GS>UAS-jou), the RNA-Seq has been performed on 7 days old flies, whose have been fed with RU486 during 7 days (starting immediately after the adult flies hatched), which is normally sufficient to well induce the transgene jouvence, and allowing enough time to jouvence to produce its effect on gene regulation. Thus, I would be surprised that the expression is not long enough, because if we look to the Figure 8, in which we have quantify by RT-qPCR, the mRNA level of several genes in the context of the re-expression of jouvence, we observe that several genes are rescued, meaning that jouvence had

already produced its effect. These RT-qPCR were performed also on 7 day-old flies. Moreover, in the Figure 8G and H (inset), the RT-qPCR have also been performed on 7 day-old flies, and we can see very nice rescued of the genes (these last flies also rely to an induction by RU486, indicating that the induction by the RU486 is very efficient).

In conclusion, this new version of Soulé et al. identifies jouvence as a snoRNA expressed in nurse cells and gut epithelium. Jouvence expression reduces enterocytes proliferation and increases lifespan in female flies. This is a novel, interesting finding which raises several questions, including the molecular mechanisms underlying this effect.

Minor comments:

Figure S3 is nice and could be inserted in the main figures

Thanks for this suggestion. However, this big Figure (with multiple panels) is considered as “secondary for the demonstration” because it show negative results consisting to exclude that other cells type express jouvence). Therefore this figure has been placed in Suppl. Figures. Moreover, as the number of Figures in the main manuscript is limited to 10, as well as the space, I prefer to keep it in Suppl. Figures. I hope that you understand my justification.

Fig 7: RNA seq data: please annotate figures so that it is immediately clear if analysis of jou in wt or overexpression.

It might have a misunderstanding, because Figure 7 refers to a comparison between WT and Deletion flies. Panels A and B refer to a comparison between Control and Deletion, considering all the genes (up- and down-regulated). However, for the sake of clarity, and to better standardize the genotype, I have renamed them in the Panel B (Heat-map).

For Panels C and D, which also refer to a comparison between Control and Deletion, it is already indicated: up-regulated (C) or down-regulated (D).

Keep figure format from figure 7 to 9: OK, but I don't completely understand the remark, because as far as I know, they are quite similar. The only difference is because we don't have any down-regulated gene in Figure 9 (overexpression). Then this is why we have just one graph as panel C. Moreover, the “Differential expression gene: log₂ fold change” of Figure 7A is not really relevant for Figure 9 because we found only 8 deregulated genes. This is why we have decided to show directly the list/Table of those 8 genes. Finally the Figure 9B is similar (Heatmap). Nevertheless, to try to be more clear, I have added “overexpression” in the Title/Head of the panel C.

Figure 8 : as supplementary:

With the addition of the two new panels (Figure 8G and H, and their inset) this figure becomes very important to show the downstream mechanism, as requested by the reviewers 2 and 4. Therefore, we think that it is relevant to keep this Figure in the main Figures.

S9: high degree of identity” instead of “homology” OK, done

S10: mis-labelling of snoRNA expression from mouse: keep the name “sno -15 “ or “sno-18” (not sno-1 or -2) while “sno-18” should be removed from the U6 hybridization.

OK, done (Thanks to the reviewer to have raised this mistake !)

S12: The qPCR experiment does not allow to evaluated the number of copies detected for each mRNA. Why would the author claim that Ctnnd2 and EPC1 are “faintly” expressed?

In RT-qPCR, the CT of the gene detected at the first dilution (generally at a dilution of 1/10, like here) is a good indicator of the level of expression of the gene.

Then, this is why we have presented the CT in the Suppl. Figure S12. As we can observe, the control genes (GAPDH for human, and 18S for mouse) have an expected CT at around 20 for GAPDH and 16 for 18S, which is a good and standard indicator. However, for the gene TEAD1, the first CT is at 27 (which is not very high but O.K.), while for the mouse, for the Ctnnd2 and EPC1, the first CT are at 33 and 29 respectively, which is very weak (almost at the limit of detection), especially for the gene Ctnnd2. Then this is why we consider that the gene Ctnnd2 and EPC1 (and particularly the gene Ctnnd2) are only faintly expressed (but they are expressed since they are detected).

REVIEWERS' COMMENTS:

Reviewer #1 (Remarks to the Author):

I was happy to accept the previous version of this manuscript and this revised manuscript contains exciting new data that identifies relevant targets in the context of ageing, so I remain enthusiastic about this study which I think should be published in its current form without further delays.

Reviewer #2 (Remarks to the Author):

This an intriguing finding and the authors deserve credit for the amount of data presented herein. The manuscript is improved by the new data

Reviewer #4 (Remarks to the Author):

In this new version, some changes have been made to improve the manuscript, notably to address the main criticism raised in the previous reviews about the mechanism by which jouvence enhances longevity. Importantly, the authors show that rescuing the expression level of two genes deregulated upon jouvence deletion, restored lifespan. This shows that jouvence effect on these genes participates to lifespan alteration. The expression levels of these two genes were not restored by jouvence re-expression in the deletion background - suggesting that their regulation is sufficient but not necessary for jou effect on survival. In the discussion, the authors mention that "genetic rescue of jou was sufficient to rescue the mRNA level of several "[genes] (p20 par. 2 l. 3 - for instance Gba1a, GstE5). Did the authors try some of these genes and found that rescuing their expression level did not suppress the effect of jou deletion on longevity? If the authors did, it would be worth mentioning.

Specific points:

The authors claim that "Jouvence is required in enterocytes / required in the gut" (title and abstract)

Since juvenile deletion does not alter the gut, it is more correct to state that juvenile is expressed/acts in the gut or is gut-specific.

p3: "the box C/D comprising the multiple "U" snoRNAs : misleading: U1, U2, U4, U5 , U6 and U7 are not snoRNAs.

Also, dysplasia and hyperplasia are two different things so the proper term but not hyperplasia/dysplasia should be used.

Transcriptomic analysis on juvenile over-expression: are some of the genes found also de-regulated upon juvenile deletion (previous study)?

Also, given the small number of up-regulated genes, I am surprised that the "pathway enrichment" is significant – for instance, from the list in F9A, I don't see any candidate related to spliceosome. In fact, all corrected p-values in the sup. Table are $>0,05$ (despite the q-value). Do the authors discussed this with statisticians?

Figure legends for RT-qPCR data need to be more descriptive, even if the information are stated in the Mat&Meth. In Fig 4, 8 and S6 please indicate on the figure (or at least in the legend) that mRNA values were normalized to that of RP49. Also indicate the number of animals used and assay repetition.

Also, it would be helpful to homogenize the qPCR data throughout the paper, from Figure 4-8 to supp. Figure 12 and 13.

Figure 5: Please indicate in the legend that light colors are for wt and dark for deletion background. Indicate in the legend that the number in each graph represent the number of animals assayed (n).

F: what is n=?

Figure 7 B and 9B: I don't understand the color code: if the left column is for mRNA levels in controls, then shouldn't all the column be in white (0)? Explain what the "fold change" refers to if they don't represent the fold-change as compared to their expression level in the controls.

Figure 9B: how come FBgn0266430 is white in “no RU” and “RU”? Doesn't its expression level change?

Indicating name of genes next to their FB number would help.

Reporting summary: OK

Source data file: Data files for survival curves are careful and complete, as well as those for transcriptomic analysis.

Gif-sur-Yvette, December 11th, 2019

Jean-René Martin
Institut des Neurosciences Paris-Saclay (Neuro-PSI)
UMR-9197, CNRS/Université Paris Sud,
Team: Functional Brain Imaging and Behavior (FBIB)
1 Avenue de la Terrasse (Bat. 32/33)
91198, Gif-sur-Yvette, France
Tel: (33) 01.69.82.41.80 / Fax: (33) 01.69.82.34.47
e-mail: jean-rene.martin@inaf.cnrs-gif.fr

Manuscript number: NCOMMS-18-15686C
jouvence* a new small nucleolar RNA required in the gut extends lifespan in *Drosophila

REVIEWERS' COMMENTS:

Reviewer #4 (Remarks to the Author):

In this new version, some changes have been made to improve the manuscript, notably to address the main criticism raised in the previous reviews about the mechanism by which *jouvence* enhances longevity. Importantly, the authors show that rescuing the expression level of two genes deregulated upon *jouvence* deletion, restored lifespan. This shows that *jouvence* effect on these genes participates to lifespan alteration. The expression levels of these two genes were not restored by *jouvence* re-expression in the deletion background - suggesting that their regulation is sufficient but not necessary for *jou* effect on survival. In the discussion, the authors mention that “genetic rescue of *jou* was sufficient to rescue the mRNA level of several “[genes] (p20 par. 2 l. 3 - for instance *Gba1a*, *GstE5*). Did the authors try some of these genes and found that rescuing their expression level did not suppress the effect of *jou* deletion on longevity? If the authors did, it would be worth mentioning.

Yes, in meanwhile, we have analyzed the *Gba1a* gene (*Del,Myo-Gal4 >UAS-Gba1a-RNAi*), but this last did not restore the longevity of the flies.

A sentence has been added, in the Discussion section, page 25 as follows: “However, these two rescues of longevity is not a systematic rule for all of deregulated genes, since we have also analysed the *Gba1a* gene (*Del,Myo-Gal4 >UAS-Gba1a-RNAi*), but this last did not restore the longevity of the flies”.

Specific points:

The authors claim that “*Jouvence* is required in enterocytes / required in the gut” (title and abstract)

Since *jouvence* deletion does not alter the gut, it is more correct to state that *jouvence* is expressed/acts in the gut or is gut-specific.

I think that it might have a misunderstanding from the reviewer 4, since we have shown, in Figure 5, that the *jouvence* deleted (*Del*) flies have, at the cellular levels, several gut lesions, as for instance, a higher number of enterocytes, more Delta-positive cells (meaning more ISCs), more dividing cells (anti-PH3 and EdU). Moreover, at the molecular levels, we have also

demonstrated that several genes were deregulated, specifically in the gut (Figure 8, and supported by the RNA-Seq analysis), and for some of them, the re-expression of *jouvence* in the gut is sufficient to rescue their mRNA levels. Therefore, to the view of these multiple independent results, we consider that *jouvence* is required in the gut, and therefore, we think that it is relevant to keep the Title and the Abstract as it is.

p3: “the box C/D comprising the multiple “U” snoRNAs : misleading: U1, U2, U4, U5 , U6 and U7 are not snoRNAs.

O.K., we have corrected this mistake and we have rewritten the sentence as follows:
page 3 in the middle of the page:

“The box C/D, comprising notably the multiple “U” snoRNAs (although not all “U” are necessarily snoRNAs), generally performs the 2’-O-methylation of ribosomal RNA, while...”

Also, dysplasia and hyperplasia are to different things so the proper term but not hyperplasia/dysplasia should be used.

O.K., right. We have removed the word “dysplasia” everywhere in the manuscript, and keep only the term “hyperplasia”, since it is more appropriated.

Transcriptomic analysis on *jou* over-expression: are some of the genes found also de-regulated upon *jou* deletion (previous study)?

Thanks the reviewer for this good remark. However, this checking was already done but we didn’t discuss of it, particularly because none of the 9 deregulated genes in the overexpression are found deregulated in the mutant (Del) (either in the up or down-regulated genes).

Also, given the small number of up-regulated genes, I am surprised that the “pathway enrichment” is significant – for instance, from the list in F9A, I don’t see any candidate related to spliceosome. In fact, all corrected p-values in the sup. Table are $>0,05$ (despite the q-value). Do the authors discussed this with statisticians?

First, in Fig 9A, the 9 genes identified by the RNA-Seq are statistically different ($p < 0,05$) (Notice that I have slightly modified the Figure 9a in the aim to lead it more readable: I have modified the display of the FoldChange and the padj). More particularly, the HSp70Bb gene (FBgn0013278) is importantly increased (FoldChange of 92,48), with a very strong p-value adjusted (padj) ($5,37 \times 10^{-12}$). However, and I agree, that the 8 remaining genes are only slightly modified.

For the spliceosome, and for the three other identified pathways, you have to refer to the Table S10, which presents the KEGG analysis pathways. For the 4 presented (and affected) pathways displayed in Figure 9c, all of them rely to the high deregulation of the HSp70Bb gene (FBgn0013278). Obviously, the high deregulation of this chaperone protein affects these 4 pathways (as calculated/revealed by the KEGG analysis, which is a well-recognized approach to identify the pathways issued from the RNA-Seq data analysis). Nevertheless, I agree with the reviewer No 4 that further analysis will be require to experimentally demonstrate such involvement in the spliceosome.

Figure legends for RT-qPCR data need to be more descriptive, even if the information are stated in the Mat&Meth. In Fig 4, 8 and S6 please indicate on the figure (or at least in the legend) that mRNA values were normalized to that of RP49. Also indicate the number of animals used and assay repetition.

O.K., We have added the missing information.

Also, it would be helpful to homogenize the qPCR data throughout the paper, from Figure 4-8 to supp. Figure 12 and 13.

In Figure 4 and 8, we have expressed and displayed the results as a $\Delta\Delta CT$, since we have two conditions to compare (control versus overexpression in Figure 4, or control versus the targeted expression of a RNAi in Figure 8), and for each of them, they are also reported to their internal control, the gene rp49. However, for the Figure S12 and S13, the displayed results correspond to ΔCT only, since mRNA level of the gene is compared only to the level of its internal control gene: GAPDH for the Figure S12, and rp49 for the Figure S13. In other words, in this last two Figures, we can't express the $\Delta\Delta CT$ since we don't have them. This is just a regular rule of qPCR. In other words, as for instance in Figure S12, we can't compare two conditions since we don't have them (for example, we don't have a mutant for this gene).

Figure 5: Please indicate in the legend that light colors are for wt and dark for deletion background.

OK., but this was already written in the first two lines of the Figure legends, as follows:

(a) Number of enterocytes (ECs) in young (7day-old: light and dark blue) and aged (40 day-old: light and dark red) flies (light = in WT and dark = in deletion).

Indicate in the legend that the number in each graph represent the number of animals assayed (n).

O.K, this was missing. We have added it in the Figure legend.

F: what is n=? (n= 2) (we have added it in the Figure legend).

Figure 7B and 9B: I don't understand the color code: if the left column is for mRNA levels in controls, then shouldn't all the column be in white (0)?

The Cluster Analysis of Gene Expression Differences is used to classify genes with similar expression levels under various experimental conditions (control versus deletion). By clustering genes with similar expression levels, it is possible to discern unknown functions of previously characterized genes or functions of unknown genes. In hierarchical clustering, areas of different colors denote different groups (clusters) of genes, and genes within each cluster may have similar functions or take part in the same biological process.

In other words, this is just a graphical "snapshot" to temptingly resume and represent the 633 deregulated genes.

In brief, Red denotes genes with high expression levels, and Blue denotes genes with low expression levels. The color range from red to blue represents the value from large to small expression. This is why we don't observe a real "white" but rather a weak pink color.

Explain what the “fold change” refers to if they don’t represent the fold-change as compared to their expression level in the controls.

More precisely, the fold change displayed corresponds, as it is written in the Figure 9, to the $\log_2\text{FoldChange}$ as it is usually and directly displayed by the RNA-Seq output data. To simplify and for more clarity, I have converted it in direct number (Fold Change) and I have modified the Figure 9 accordingly (then, this is more easy and straightforward to read). (Notice that $\log_2\text{FoldChange}$ is the log in basis 2: which is the displayed rule for RNA-Seq).

Figure 9B: how come FBgn0266430 is white in “no RU” and “RU”? Doesn’t its expression level change? see explanation above.

Indicating name of genes next to their FB number would help.

I don’t think that it is necessary, since it is written just above in the panel A.

Reporting summary: OK

Source data file: Data files for survival curves are careful and complete, as well as those for transcriptomic analysis.